ecology, immunology, physiology

ecoimmunology, immune function, physiological trade-offs, red crossbill, seasonality, annual cycle

**Author for correspondence:**
Elizabeth M. Schultz
e-mail: schultze@wittenberg.edu

†Present Address: Department of Biology, Wittenberg University, Springfield, Ohio, USA.

# Patterns of annual and seasonal immune investment in a temporal reproductive opportunist

Elizabeth M. Schultz[1,†], Christian E. Gunning[3,4], Jamie M. Cornelius[5], Dustin G. Reichard[6], Kirk C. Klasing[2] and Thomas P. Hahn[1]

[1]Department of Neurobiology, Physiology, and Behaviour, and [2]Department of Animal Science, University of California Davis, Davis, CA, USA
[3]Odum School of Ecology, University of Georgia, Athens, GA, USA
[4]Affiliated Scholar, Department of Mathematics and Statistics, Kenyon College, Gambier, OH, USA
[5]Department of Integrative Biology, Oregon State University, Corvallis, OR, USA
[6]Department of Zoology, Ohio Wesleyan University, Delaware, OH, USA

EMS, 0000-0003-0352-6967; CEG, 0000-0001-6403-6553; JMC, 0000-0002-6334-7277; DGR, 0000-0002-1219-9219

Historically, investigations of how organismal investments in immunity fluctuate in response to environmental and physiological changes have focused on seasonally breeding organisms that confine reproduction to seasons with relatively unchallenging environmental conditions and abundant resources. The red crossbill, *Loxia curvirostra*, is a songbird that can breed opportunistically if conifer seeds are abundant, on both short, cold, and long, warm days, providing an ideal system to investigate environmental and reproductive effects on immunity. In this study, we measured inter- and intra-annual variation in complement, natural antibodies, PIT54 and leucocytes in crossbills across four summers (2010–2013) and multiple seasons within 1 year (summer 2011–spring 2012). Overall, we observed substantial changes in crossbill immune investment among summers, with interannual variation driven largely by food resources, while variation across multiple seasons within a single cone year was less pronounced and lacked a dominant predictor of immune investment. However, we found weak evidence that physiological processes (e.g. reproductive condition, moult) or abiotic factors (e.g. temperature, precipitation) affect immune investment. Collectively, this study suggests that a reproductively flexible organism may be able to invest in both reproduction and survival-related processes, potentially by exploiting rich patches with abundant resources. More broadly, these results emphasize the need for more longitudinal studies of trade-offs associated with immune investment.

## 1. Introduction

Many temperate, terrestrial organisms experience extensive seasonal variation in weather, disease exposure and resource availability across the annual cycle. In turn, natural selection favours strategies that balance seasonal allocation to both reproduction and self-maintenance to maximize fitness [1]. Investment in immune function promotes survival by minimizing deleterious effects of pathogens and disease [2]. However, the energy and opportunity costs involved in maintaining immunity can be high [3,4]. Empirical data suggest that changing environmental conditions (e.g. pathogens, resource availability) strongly influence allocation to immunity [5,6]. If varying environmental conditions most strongly influence immune allocation, then investment in immunity would vary significantly both within and between years according to prevailing conditions (*sensu* Hegemann *et al.* [7]). However, organisms may also modulate immunity in direct response to an energy trade-off with competing processes such as

reproduction, migration, or plumage/pelage moult, resulting in predictable seasonal or inter-annual patterns of immune investment (*sensu* Hegemann *et al.* [7]).

Effects of varying environmental conditions and competing physiological processes on immune investment are not mutually exclusive, and our ability to quantify the relative contributions of these effects has been limited by both experimental methodology and study systems. Previous research on seasonal variation in immunity has focused on small mammals [8], while the majority of studies on birds have focused on single life-history stages of the annual cycle [5]. Though limited in scope, this research shows that immune investments decrease during reproduction (e.g. [9,10]), moult [11,12], migration [13,14] and winter [15,16], with notable exceptions [5,17]. Fewer studies, however, have examined modulations in immunity across multiple annual cycle stages. Some components of immunity e.g. microbial-killing ability, were reduced during breeding in house sparrows (*Passer domesticus*) [18], yet other components such as complement activity, natural antibody levels, total immunoglobulin levels and antibody response to multiple foreign antigens were higher during breeding in house sparrows [18], great tits (*Parus major*) [19] and skylarks (*Alauda arvensis*) [7], relative to birds caught during moult and winter.

Detailed observations across multiple years can assist in disentangling the effects of environmental and physiological processes on immune investment. To date, a few studies have demonstrated significant interannual variation in metrics of constitutive immunity (complement activity, natural antibody and haptoglobin levels) in skylarks [7] and seven species of Galápagos finches [20], suggesting the importance of inter-annual variation in environmental conditions. However, these studies of long-term temporal variation did not examine potential environmental predictors. Multi-year studies of equatorial species quantifying environmental variation demonstrate the complexity of immune-environment interactions. For example, precipitation and ambient temperature were not related to haptoglobin, complement or natural antibodies in red-capped (*Calandrella cinereal)* or rufous-naped larks (*Mirafra africana*) [21], whereas precipitation was related to higher microbial killing capacity in neotropical house wrens (*Troglodytes aedon*) [22], but lower haptoglobin, ovo-transferrin, complement and natural antibodies in common bulbuls *Pycnonotus barbatus* [23].

Most previous research into potential drivers of immune investment has focused on seasonal breeders that perform their most demanding physiological processes during seasonal periods of high resource availability and more benign environmental conditions. This close correspondence between resource availability and physiological demands complicates efforts at quantifying the relative importance of environmental and physiological factors on immune investment [5,6]. As such, more studies are needed on free-living vertebrates across multiple seasons within a year, and across multiple years, to better understand the factors underlying seasonal differences in immunity. Furthermore, studies of organisms exhibiting reproduction that is facultative across a wide range of environmental conditions permits a more direct assessment of how physiological demands and environmental fluctuations influence the evolution of life history-related investments in immunity.

Here, we present a multiannual study of a songbird, the red crossbill (*Loxia curvirostra*), that breeds both in summer and winter if food (i.e. conifer seeds) is sufficiently abundant [24].

We focused on constitutive immunity, an important first-line of defence against invading pathogens, because its consistent production costs may underlie physiological trade-offs [5,25]. Specifically, we measured inter- and intra-annual variation in complement, natural antibodies, acute-phase protein concentration (PIT54), and the relative abundances of circulating leucocytes in free-living crossbills across four consecutive summers (2010–2013) and across multiple seasons within one year (summer 2011—spring 2012) in Wyoming, USA. We also measured a range of environmental (e.g. ambient temperature and food availability), physiological (e.g. breeding and integument moult) and other covariates that previous work has suggested may affect one or more of these immune parameters (e.g. sex and age, reviewed in Adelman [17]).

In contrast to previous studies, we used a two-tiered modelling approach to assess the relationship between immune parameters and potential biotic and abiotic covariates. We first constructed a suite of random forest models (RFMs), which is a non-parametric approach that carries fewer assumptions than traditional linear models (LMs), while allowing for nonlinear effects and higher-order interactions [26]. These RFMs identified the specific covariates that warranted further exploration as putative predictors of immune investment (i.e. variable or 'feature' selection) [27,28]. For each immune measure, we then constructed separate LMs containing the covariates selected above. This approach allowed us to focus on the most relevant covariates in a familiar LM framework, while still providing a comprehensive descriptive analysis of all available data [29].

## 2. Material and methods

### (a) Field methods

#### (i) Study species and site

Red crossbills (*L. curvirostra*) are nomadic, reproductively flexible passerines that eat mostly conifer seeds, the availability of which varies dramatically in space and time [24,30,31]. In years with abundant cone crops, crossbills can breed from late summer to the subsequent spring, with a hiatus in mid to late autumn for moult, and can have multiple broods per year, despite thermal challenges and short days in some seasons [32–34]. Crossbills are categorized as 'seasonal opportunists;' while they are more behaviourally flexible than seasonal migrants, they also exhibit seasonal cycles of migratory physiology, reproduction and moult that are not controlled proximally only by variation in food supply [35–38].

Data were collected from free-living red crossbills from 2010 to 2013 around Grand Teton National Park, Wyoming, USA (43° 45′ N, 110° 39′ W), a montane temperate-zone environment with large seasonal fluctuations in day length, food availability, temperature and precipitation (figure 1*a*; electronic supplementary material, figures S2 and S4). Crossbills were sexed and aged using plumage and skull as described in Pyle [39]. At this site, red crossbill abundance fluctuates from year-to-year and is related to the cone crop size on dominant conifers [34,40]. The dominant conifers used by crossbills here are lodgepole pine (*Pinus contorta*), Douglas-fir (*Pseudotsuga menziesii*), Engelmann spruce (*Picea engelmannii*) and blue spruce (*Picea pungens*). Ten described vocal 'types' of red crossbills can be categorized into four classes by body size and bill morphology [41–43]. These morphological differences among types optimize their foraging efficiency on specific conifer taxa [41,42,44]. This study presents data from vocal types 2, 3, 4 and 5 (electronic supplementary material, table S2).

*Proc. R. Soc. B* **287**: 20192993

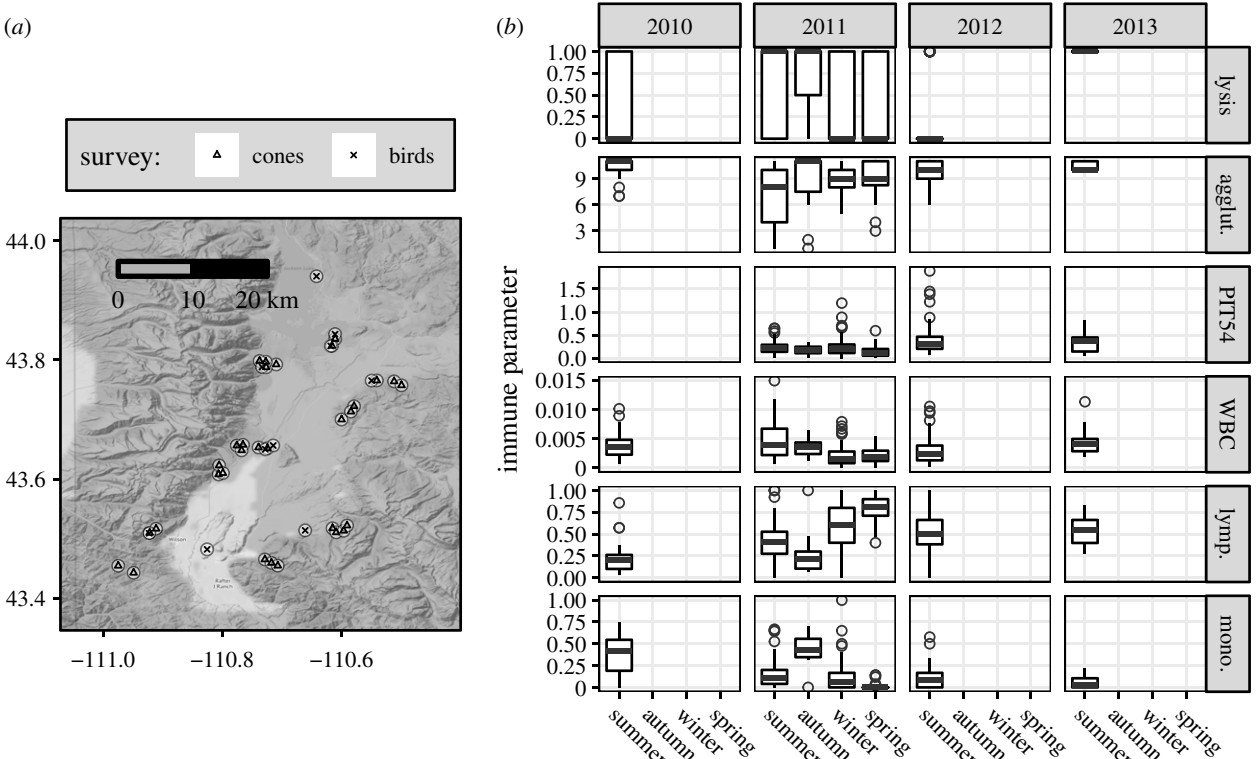

**Figure 1.** (*a*) Map of survey sites (Teton County, Wyoming, USA). (*b*) Overview of immune parameter observations. Column labels show cone year (1 June of the current year through to 31 May of the subsequent year). Crossbills were sampled during the summer season of every cone year and were sampled in each season during cone year 2011 (1 June 2011–31 May 2012). Responses: lysis (positive hemolysis score: $F = 0$, $T = 1$), agglut. (agglutination score), PIT54 (mg ml$^{-1}$), WBC (proportion leucocytes/erythrocytes), lymp. (proportion lymphocytes/leucocytes), mono. (proportion monocytes/leucocytes).

### (ii) Delineation of season and cone year

Sampling periods were categorized into seasons: birds caught in the summer were caught from 23 June to 12 September, autumn from 25 to 30 October, winter from 1 to 11 March and spring from 3 to 9 May. Sample sizes per year and season appear in the electronic supplementary material, table S1. A 'cone year' coincides with the cone development occurring between approximately 1 June of one year until the following spring when old cones are depleted or new cones start developing [45] (see below).

### (iii) Capture methods and blood sampling

Crossbills were lured into mist nets with live caged decoys and/or playback. Approximately 300 µl of blood per bird was collected from the brachial vein into heparinized microhematocrit capillary tubes. This collection occurred between 7.00 and 20.00 h with a median elapsed time from capture to sampling of 3.73 min (maximum of 60 min) to minimize potential effects of rising glucocorticoids [46]. Blood samples were held on ice for no more than 7 h before centrifuging (10 min at 10 000 rpm, IEC clinical centrifuge) and separated plasma was stored at −20°C until immune assays were performed. Per cent packed cell volume (hematocrit) was measured in all birds except those captured during summer 2010.

### (b) Immune assays

#### (i) Complement and natural antibodies (lysis and agglutination)

The protocol described in Matson *et al.* [47], was used to measure plasma complement activity and non-specific natural antibodies via red blood cell lysis and agglutination, respectively (figure 1*b*). Partial lysis and agglutination were indicated by half scores. Samples were scored blind to sampling date and randomized across plates by one observer (E.M.S.). A positive standard (chicken plasma) was run on all plates in duplicate. Ten millilitres

of plasma was used owing to small blood volumes and reagent volumes were adjusted accordingly. Samples were run in five batches: December 2010 ($n = 29$), December 2011 ($n = 67$), May 2012 ($n = 123$), October 2012 ($n = 98$) and June 2014 ($n = 13$). Repeated freeze–thaw cycles do not affect assay results [48]. The average inter-plate variation (% coefficient of variation; CV) was 5.04% (lysis) and 0.79% (agglutination). Owing to the abundance of lysis scores of zero in our dataset (60.7% zero scores; non-zero scores ranged from 0.4 to 5), we assigned individuals a 0 or 1 score, where 1 was any non-zero lysis score.

#### (ii) Haptoglobin (PIT54)

To quantify plasma PIT54 concentrations, a colorimetric assay kit (TP801; Tri-Delta Diagnostics, NJ, USA) was used (figure 1*b*). To accommodate small blood volumes, 5 µl was used and all reagents were adjusted accordingly. Because additional plasma was needed to optimize the haemolysis-haemagglutination assay, we did not measure haptoglobin values in 2010. Samples were run simultaneously by E.M.S. in October 2014 and randomized across seven plates. Mean inter-assay CV was 5.4% and mean intra-assay CV was 5.6%.

#### (iii) Circulating cellular immunity (WBC)

To identify the quantity and type of leucocytes (lymphocytes, heterophils, monocytes, eosinophils and basophils), a drop of whole blood was spread onto a slide, air-dried, fixed with 100% methanol and stained with Wright–Giemsa (Cambridge Diagnostic Camco Stain Pack). The number and type of leucocytes under 1000× magnification were scored by two observers using the methodology outlined in Campbell [49]: E.M.S. scored all seasons except for summer 2012 which was done by D. Jaul, who was trained by and calibrated against E.M.S. using a subset of the same slides to validate cell identification and quantification. Leucocytes were detected across 100 microscope 'fields' or approximately 10 000

erythrocytes and reported as the number of leucocytes per number of fields scored (figure 1b). We also calculated the heterophil to lymphocyte ratio, and the relative proportion of each leucocyte type [50]. Eosinophil, heterophil, basophil and heterophil to lymphocyte ratio models had low (marginal) explanatory value ($R^2 < 0.05$); we only discuss results from overall leucocytes (WBC), lymphocytes (lymp.) and monocytes (mono.).

## (c) Physiological measures

### (i) Reproductive measures

Male cloacal protuberance (CP) length was measured with dial callipers from abdomen to cloacal tip; this measure correlates to testis length in free-living male red crossbills [35]. For females, the brood patch (BP) was scored as 0 (a dry, fully feathered breast), 1 (loss of feathers, no vascularization), 2 (loss of feathers with mild oedema and/or vascularization), 3 (loss of feathers, full oedema/vascularization), or 4 (bare and wrinkly breast, post full oedema); BP scores of greater than 0 significantly predict crossbill ovary condition [35].

### (ii) Plumage moult intensity

Pre-basic moult occurs seasonally in red crossbills (June–November) and may be arrested during summer breeding [24,37]. The primary flight feathers grow sequentially from wrist to wingtip [51], and the number of actively growing feathers was defined as primary or flight feather moult intensity. Contour feather (body) moult was scored by surveying the entire body and rated on a scale of 0 (no pins or sheaths present), 1 (light: few pins, growing or sheathed feathers in one tract), 2 (medium: approx. 10–20 pins, growing or sheathed feathers in multiple tracts), and 3 (heavy: many pins, growing/sheathed feathers across multiple feather tracts) [51].

### (iii) Mass, fat and structural measures

Body mass was measured to the nearest 0.25 g with a Pesola spring scale and furcular and abdominal fat were scored on a scale from 0 (no fat) to 5 (bulging) [52,53]. Tarsometatarsus was measured using dial callipers (in millimetres) and body condition was calculated by performing linear regressions of mass by tarsus length and calculating residuals.

## (d) Environmental measures

### (i) Cone crop

To evaluate the availability of conifer food sources in the area (lodgepole pine, Douglas-fir, Engelmann spruce and blue spruce), one experienced observer (T.P.H.) visited 12 distinct, long-term point-count sites between July and September of each year (electronic supplementary material, figure S4). These sites were established in 2006, and at each annual visit, 10–20 mature trees of each species present within 50 m of the point-count site were assigned a cone abundance index (USFS 1994) ranging from 0–5 (0 no cones) to 5 (abundant cones on a cone-bearing section of tree) [40]. Seed supply and crossbill foraging intensity is greatest in summer and early autumn when cones are ripening and beginning to open [32]. Seed abundance declines in winter and spring owing to seed shedding and predation [54].

### (ii) Local weather conditions

For each day of bird capture, 24 h precipitation amounts (mm) and daily maximum and minimum temperature (degrees Celsius) were accessed from the National Oceanic & Atmospheric Administration (NOAA) National Climate Data Centre website, using the weather station MOOSE 1 NNE, Wyoming, USA (elevation: 1970.84 m, latitude: 43.662° N longitude: 110.712° W) (electronic supplementary material, figure S2). By subtracting the daily minimum ($T_{min}$) from maximum temperatures ($T_{max}$), we calculated the temperature difference ($T_{diff}$) for each capture day to calculate diel temperature range, which can vary substantially in a montane environment. Minimum and maximum daily temperature were highly correlated; we thus omitted maximum daily temperature and only included $T_{min}$, which in this climate would have a greater impact on thermoregulatory demand, and $T_{diff}$ for subsequent analyses.

## (e) Statistical analyses

### (i) Overview

All analysis was conducted in R v. 3. 6.1 [55]. We constructed a set of statistical models for each of six separate immune parameters (lysis, agglut., PIT54, WBC, lymp., mono.). We use separate models for observations across multiple seasons within a single cone year, and summer across multiple cone years (season, year, resp.). Observations from summer 2011 were included in both models. Immune measures were not highly correlated ($\rho < 0.2$), except for mono. and lymp. ($\rho = -0.6$) (electronic supplementary material, figure S8). Model predictors included environmental, physiological, intrinsic and sampling-related covariates (electronic supplementary material, table S3). Environmental covariates included the daily minimum temperature, diel temperature range and precipitation. Physiological covariates included CP length/ BP score, primary and contour feather moult intensity, haematocrit score, residual body mass score and composite fat score. Intrinsic covariates included age, sex and vocal type. Sampling-related covariates included capture location, time of day, and time elapsed between capture and blood sampling. We first used RFMs for variable selection, and then constructed LMs for statistical inference using the variables identified by the RFMs [56].

### (ii) Statistical models

Owing to data limitations, we first separated data into two groups based on sampling timing: yearly (summer, cone years 2010–2013) and seasonal (summer and fall 2011, winter and spring 2012, i.e. cone year 2011); see 'Delineation of season' in Methods for date ranges. Each yearly and seasonal model included sampling period (cone year or season, respectively) as a predictor.

For each combination of sampling period and immune measure, we constructed an RFM using all measured covariates except for haematocrit, capture location, capture time of day and vocal type (electronic supplementary material, table S3). Haematocrit was excluded from further consideration owing to missing observations, capture location and time of day were excluded owing to strong association with season, and vocal type was excluded owing to unbalanced sample sizes among sampling groups (electronic supplementary material, tables S2 and S3). Each RFM comprises 2000 unbiased conditional inference trees (R package *party*, [57–59]). In summary, individual trees are constructed from a bootstrapped sub-sample of the original data and tree accuracy is evaluated from the remaining out-of-sample (out-of-bag) observations. RFM variable importance is then computed as the mean decrease in accuracy (over all trees) when each covariate in a tree's out-of-sample observations are permuted (i.e. randomized) [26].

For each model, we identified covariates with negative variable importance and removed these from future consideration. We then re-fitted the RFM using all covariates with positive variable importance. We repeated this iterative process a total of three times. As such, the remaining predictor variables demonstrated a consistent statistical association with the given immune parameter predicted by the RFM.

For each combination of sampling period and immune measure, we then constructed an LM that included the covariates selected above, i.e. those with positive variable importance in the final RFM. A logistic generalized linear model (GLM) was used to model lysis (0= no haemolysis, 1= non-zero haemolysis), which we refer to simply as the lysis LM. For each LM, we report

**Table 1.** Goodness-of-fit (adjusted $R^2$) for each model of immune parameter response (lysis, GLM; all others, LM). (Model $p$-values were computed by an $F$-test except for the lysis GLM, where a likelihood ratio test was used. For clarity, only models with $R^2 > 0.1$ are further considered.)

| period | response | sample size | $R^2$ | $p$-value | period | response | sample size | $R^2$ | $p$-value |
|---|---|---|---|---|---|---|---|---|---|
| year | lysis | 191 | 0.316 | $5.2 \times 10^{-14}$ | season | lysis | 180 | 0.103 | 0.00093 |
| | agglut. | 191 | 0.262 | $3.4 \times 10^{-11}$ | | agglut. | 182 | 0.089 | 0.00098 |
| | PIT54 | 157 | 0.082 | 0.0059 | | PIT54 | 170 | 0.128 | 0.00018 |
| | WBC | 175 | 0.105 | 0.0011 | | WBC | 158 | 0.277 | $1.6 \times 10^{-09}$ |
| | lymp. | 176 | 0.187 | $4.6 \times 10^{-07}$ | | lymp. | 155 | 0.210 | $8.2 \times 10^{-06}$ |
| | mono. | 177 | 0.267 | $5.63 \times 10^{-10}$ | | mono. | 157 | 0.197 | $5.2 \times 10^{-06}$ |

type II ANOVAs for each predictor. We also report an overall model $p$-value using an $F$-test (except lysis, where a likelihood ratio test was used), and an adjusted $R^2$ that was calculated using the $rsq.v$ function in the R package $rsq$. To assess the direction and magnitude of immune measures over time, we used each LM to calculate the expected marginal means (EMM) within each respective sampling period (i.e. by cone year or season) (R package $emmeans$ [60]). We also used these EMMs to contrast between time periods, e.g. using pairwise honest significance differences (HSD) with Tukey adjustments for family-wise error rate (FWER, $p < 0.05$) within each model. We also calculated EMMs of cone crop scores within each cone year (marginalized over species) and compared these yearly EMMs to the EMMs of each immune variable within a cone year.

## 3. Results

### (a) Patterns of variation across cone years (summers 2010, 2011, 2012, 2013)

LMs predicting variation in immune parameters across the summer of multiple cone years were highly significant ($p < 0.001$): for complement (lysis), natural antibodies (agglut.), PIT54, leucocytes/erythrocytes (WBC), lymphocytes (lymp.) and monocytes (mono.), adjusted $R^2$ ranged from approximately 0.082 (PIT54) to 0.32 (lysis) (table 1).

Cone year had positive variable importance and was identified as a significant predictor in all RFMs and LMs, respectively (please see referenced tables for exact $p$-values; electronic supplementary material, figure S1 and tables S7–12). An examination of the EMM of each immune parameter by cone year showed that complement was significantly higher in 2011 than in 2010 and 2012, though not significantly different from 2013 (figure 2$a$; electronic supplementary material, table S4). Similarly, leucocytes/erythrocytes (WBC) were significantly higher in 2011 than in 2012, though not significantly different from 2013 and 2010 (figure 2$a$; electronic supplementary material, table S4). On the other hand, natural antibodies and PIT54 were, on average, lowest in 2011 and highest in 2010, 2012 and 2013. Overall, lymphocytes and monocytes were inversely correlated ($\rho = -0.64$, electronic supplementary material, figure S8). Compared across cone year, lymphocytes were lowest in 2010 and higher in 2011, 2012 and 2013, whereas monocytes were highest in 2010 and lowest in 2011, 2012 and 2013 (figure 2$a$; electronic supplementary material, table S4).

Annual cone crop scores varied significantly among tree species (electronic supplementary material, figure S5), while cone crop scores averaged across species varied substantially across cone years (electronic supplementary material, figure S4). Focusing on mean yearly cone crop scores, we found

that complement and leucocytes/erythrocytes were generally higher in cone years with higher cone crops (2011, 2013) and lower in cone years with smaller crops (2010, 2012), while our data suggest an inverse relationship between cone crop scores and both natural antibodies and PIT54 (figure 2$a$; electronic supplementary material, figure S6). On the other hand, monocytes and lymphocytes show no consistent relationship with cone crop scores, though appear inversely related across cone years (figure 2$a$; electronic supplementary material, figure S6).

Capture duration (capture.dur.) was the only other variable with positive variable importance and identified as a significant predictor in both RFMs and LMs, respectively (electronic supplementary material, figure S1 and table S6). Specifically, time elapsed between capture and blood sampling was negatively related to leucocytes/erythrocytes (electronic supplementary material, table S6).

### (b) Patterns of variation across seasons (summer, autumn, winter, spring of cone year 2011)

Season had the highest variable importance in all RFMs except for PIT54, where body condition (R.mass) was highest, and natural antibodies, where minimum daily temperature ($T_{\min}$) was the highest (electronic supplementary material, figure S1). As with yearly models, all LMs predicting immune parameters across the seasons of cone year 2011 were highly significant ($R^2 > 0.1$, $p < 0.01$), with the adjusted $R^2$ ranging from approximately 0.09 (agglut.) to 0.28 (WBC) (table 1).

When examining seasonal EMMs of immune parameters, monocytes and lymphocytes displayed significant differences between seasons (electronic supplementary material, table S5). Lymphocytes were highest in spring and lowest in autumn, while monocytes were lowest in spring and summer and highest in autumn, again suggesting an inverse relationship between the two (figure 2$b$; electronic supplementary material, table S5). On the other hand, complement, natural antibodies, leucocytes/erythrocytes and PIT54 showed no significant differences among seasons (figure 2$b$; electronic supplementary material, table S5).

Several environmental and capture-related variables had positive variable importance in RFMs and were identified as significant predictors of variation in immune parameters in LMs (electronic supplementary material, figure S1 and table S6). Minimum daily temperature ($T_{\min}$), diel temperature range ($T_{\text{diff}}$) and precipitation (precip.) were identified as significant, positive predictors of variation in leucocytes/erythrocytes (electronic supplementary material, tables S6 and S11). In addition, precipitation (precip.) and body condition (R.mass) were significantly, positively related to PIT54

*Proc. R. Soc. B* **287**: 20192993

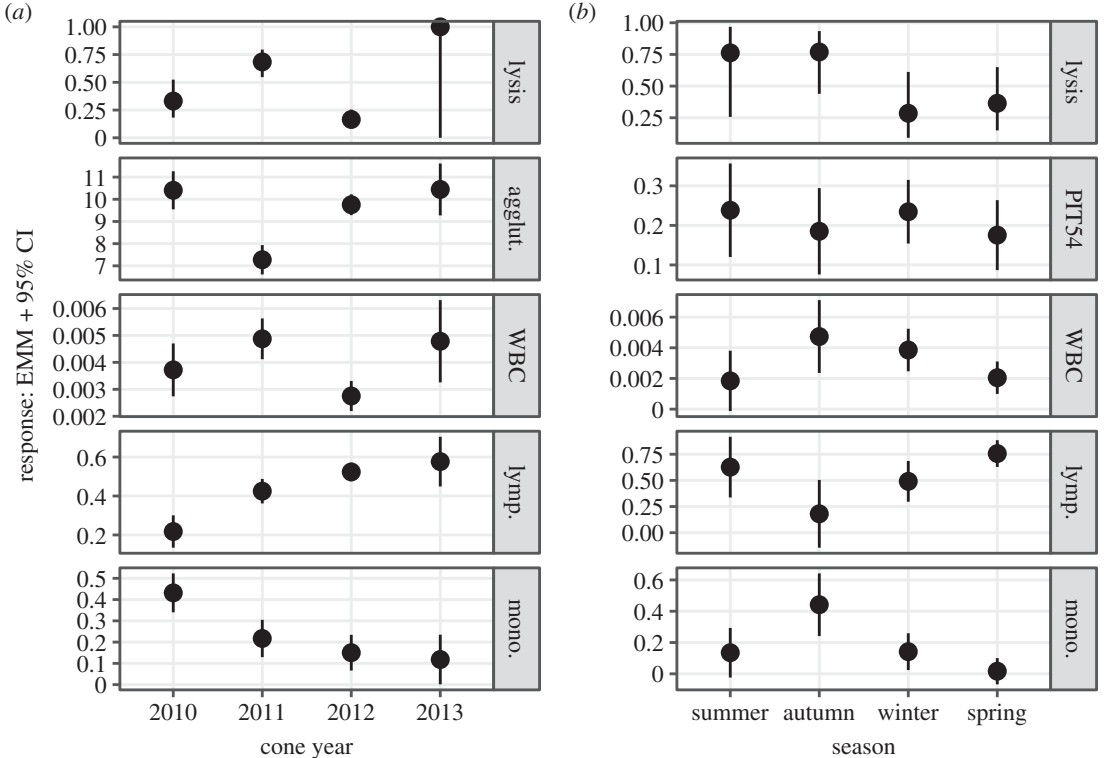

**Figure 2.** Expected marginal mean (EMM) and 95% confidence interval (CI) of each model response by time period: (*a*) cone year (all years, summer only), (*b*) season (all seasons, cone year 2011 only). See also table 1.

(electronic supplementary material, tables S6 and S12). Finally, capture duration was significantly, negatively related to lymphocytes (electronic supplementary material, tables S6 and S9).

## (c) Variables warranting further study

For both yearly and seasonal models, covariates with consistently low variable importance included reproductive condition (CP/BP), sex and age (electronic supplementary material, figure S1). A number of covariates exhibited positive variable importance in RFMs (electronic supplementary material, figure S1), and yet were not statistically significant predictors in subsequent LMs, despite these LMs being highly significant (table 1; electronic supplementary material, table S7–12). For example, the number of actively growing flight feathers (Ff) was included in statistically significant yearly LMs predicting variation in complement and leucocytes/erythrocytes and in seasonal LMs predicting complement, leucocytes/erythrocytes, lymphocytes and monocytes (electronic supplementary material, tables S7 and S9–11). Similarly, diel temperature range was included in yearly LMs predicting variation in natural antibodies, leucocytes/erythrocytes and lymphocytes, and in seasonal LMs for complement, lymphocytes, monocytes and leucocytes/erythrocytes (electronic supplementary material, tables S7–11). Other covariates that exhibited positive variable importance but were not statistically significant in multiple LMs included minimum daily temperature ($T_{min}$) and contour feather moult intensity (body.molt) (electronic supplementary material, figure S1 and tables S7–12).

## 4. Discussion

Here, we provide analyses based on detailed observations of crossbill immune parameters and a suite of physiological and environmental covariates across four consecutive summers

(2010–2013), and across a full 12 months (2011–2012). Our goal was to provide a 'comprehensive descriptive study' [29] that highlights both negative and positive evidence for the ecological determinants of immune function. We employ a two-tiered approach, where non-parametric RFMs provide evidence against covariates (i.e. variable importance less than or equal to zero), and LMs quantify positive evidence, if any, for the influence of remaining covariates. Overall, we observed substantial changes in crossbill immune investments among summers across four years, with interannual variation driven largely by food resources, while seasonal variation was less pronounced and lacked a dominant predictor.

Although 'costly' physiological processes such as reproduction and moult can affect immune investment [9,11,61], our data only weakly supported this prediction. Flight feather moult (Ff), plumage moult intensity (body.molt) and reproductive measures (CP/BP) were all selected by RFMs for further consideration (electronic supplementary material, figure S1), yet of these, only flight feather moult displayed a marginally significant, positive relationship with complement (electronic supplementary material, table S6). In temperate-zone breeding, Northern Hemisphere birds, flight and contour moult are generally heaviest during late summer/early autumn (electronic supplementary material, figure S7, [51]), which in crossbills corresponded to the lowest and highest lymphocyte and monocyte proportions, respectively (figure 2*b*; electronic supplementary material, table S5). Elevated monocyte concentration during moult has also been documented in red knots (*Calidris canutus*) [62]. Given that feathers erupting through the skin can induce dermal inflammation [63] and chemotaxis of blood monocytes that differentiate into macrophages that are integral to the inflammatory response [49], elevated monocytes during peak moult is not surprising. A modest decrease in lymphocytes during this same period

may indicate a trade-off within the immune system to favour phagocyte-mediated responses rather than cellular and antibody-based immunity. We note that sample sizes were limited during autumn 2011 (electronic supplementary material, table S1); despite substantial field effort, capturing crossbills during moult is especially difficult because more secretive behaviours accompany this stage [64].

Results from RFMs eliminated our reproductive measure (CP/BP) as a reliable indicator of immune variation from all but two cases (electronic supplementary material, figure S1). However, while CP length and BP score are correlated with testes length and ovary condition in crossbills [35], these measures do not quantify the total energetic costs incurred throughout reproduction (e.g. egg laying, incubation, provisioning offspring) [65]. For crossbills rearing young in winter, reproduction is presumably energetically demanding owing to the intensive foraging behaviour required to meet the higher energy and thermoregulatory demands of themselves, incubating mates and offspring. However, while metabolic rate may be elevated, baseline corticosterone levels, which reflect current energy demands [66], are not higher in winter-breeding crossbills, nor is corticosterone higher in breeding crossbills in general [35], (J.M. Cornelius 2002–2009 unpublished data). Thus, despite elevated demands, crossbills probably cope with these costs because reproduction in winter primarily occurs when conifer seeds are abundant. Crossbills also may incur lower overall reproductive costs compared with other seasonal species because they do not defend territories and have relatively small clutch sizes [24], thus potentially facilitating investment in reproduction and immunity.

Crossbills rely on the cone crops of four conifer species as their primary food resources at our study site. Substantial annual variation in these cone crops (electronic supplementary material, figures S4–S5) may be driven, in part, by past climatic conditions [67,68]. This annual variation in food resources corresponded with annual variation in several immune measures. Across all four study years, complement and leucocytes/erythrocytes were higher when cone crops were larger, whereas natural antibodies and acute phase proteins (PIT54) were somewhat higher when cone crops were smaller, across three study years (electronic supplementary material, figure S6). Higher levels of natural antibodies and PIT54 but lower levels of complement and leucocytes during lower cone years may reflect a shift in immune investment strategy in response to lower food resources. While investment in the more rapid but costly PIT54 defence contradicts the energy-limitation hypothesis [69,70], this trade-off suggests that the overall costs of constitutively maintaining higher levels of protective proteins and cells could be higher than occasionally inducing an expensive inflammatory response. We do not, however, have repeated samples from individuals which would establish baseline levels, making it difficult to ascertain whether higher levels of PIT54 indicate inflammation or are within an individual's normal range. In addition, capture duration was negatively related to lymphocytes and overall leucocytes/erythrocytes, suggesting that handling time and thus corticosterone may affect leucocytes [71]. Finally, our analysis is limited by a small number of cone years, which further highlights the importance of long-term monitoring to assess organismal responses to annual cycles.

The variation in crossbill immune activity across multiple seasons within a single cone year (summer 2011–spring 2012), corresponded to the full annual cycle of one large, cumulative cone crop. Immune variation among seasons, however, was modest relative to annual variation. LMs of immune parameters within this year were highly significant for all responses (table 1). Separately, season itself was a significant predictor only of lymphocytes and monocytes (figure 2b; electronic supplementary material, tables S9 and S10). While precipitation, ambient temperature and/or diel temperature range were included in seasonal LMs predicting complement, natural antibodies, PIT54, lymphocytes and monocytes, these covariates were only identified as individually significant predictors for leucocytes/erythrocytes and PIT54 (see below) (electronic supplementary material, tables S6 and S12). It is plausible that we were unable to detect significant linear effects of these covariates on these immune variables owing to the complex or interactive effects of environmental conditions on organismal physiology, yet their inclusion in multiple, highly significant seasonal LMs (table 1; electronic supplementary material, table S7–S12) suggests a potential effect on crossbill immune investment.

Although there were no significant seasonal differences in leucocytes (figure 2b), both diel and minimum daily temperature were significant, positive predictors and tended to be highest during the summer of the 2011 cone year (electronic supplementary material, figure S3). This relationship between temperature and leucocytes may be owing to a higher probability of disease and infection during the summer months. For example, red crossbills have higher Haemoproteus infections in the late spring and early summer than other times of year [45,72], and this was significantly related to leucocyte counts [45]. While PIT54 also did not exhibit significant seasonal patterns, precipitation was a significant, positive predictor. Overall, higher precipitation levels, in the form of rain and snow, occurred more frequently in summer and winter, respectively (electronic supplementary material, figure S3). These higher precipitation levels may have created a more challenging foraging and/or thermal environment which in turn could have made infection or inflammation owing to immune activation more likely [70].

Evidence describing relationships between condition and immune investment in free-living organisms is mixed, with research suggesting no relationship between body condition and specific antibody response in mallards (Anas platyrhnchos) [73], and other work showing that elk (Cervus elaphus) in poor nutritional condition invest more in constitutive immune measures [74]. Here, however, body condition related positively to PIT54, suggesting that good condition allowed for higher PIT54, PIT54 was not reflective of immune challenge, or infection did not cause reduced body condition. PIT54 was also highest in cone year 2012, which had a small cone crop, suggesting that immune investment, particularly in the acute phase response, may be prioritized when resources are limited or conditions are challenging in order to maintain adequate defence against pathogens [75].

While our study suggests several factors that influence immune investment, some of the observed variation is probably in response to photoperiod change. For example, previous work on captive red crossbills found that total leucocyte counts and bacterial killing ability increased in response to long days [76]. In addition, our characterization of inter- and intra-annual variation in immunity is based on sampling from exclusively summer months and within one cone year, respectively. As such, we cannot fully disentangle environmental and

physiological contributors to immunity, particularly because physiological covariates like reproduction, moult and condition may vary significantly between and within years. In our study, reproduction, fat, flight feather moult, body moult and condition did vary both between and within years, although not substantially (electronic supplementary material, figure S7). Finally, we also note that our study included multiple vocal types of crossbills that could have originated from different populations and thus experienced different environments and pathogens, either prior to or after arriving at our study site. However, owing to limited sample sizes for most types within and among sampling periods, we are unable to assess the influence of vocal type here.

The variation in immune parameters found in this study highlights the importance of sampling across multiple years and seasons in order to draw robust conclusions about the seasonality of immunity in wild organisms. Overall, this study supports previous findings that birds seasonally modulate investment in immunity [18,19], investment in immunity changes between years e.g. [7,20], and this variation is partially explained by environmental conditions e.g. [22,23]. While we cannot unambiguously identify the causal factors driving the observed variation in immune parameters, annual cone crop scores nonetheless suggest that food resources may explain some of the observed interannual variation, particularly for complement and total leucocytes (electronic supplementary material, figure S6). We also observe weak relationships between several measures of immune investment and ambient temperature, precipitation and plumage moult. On the other hand, RFMs provide negative evidence against the effect of reproduction (CP/BP) on any immune parameters except for PIT54 and lymphocytes (electronic supplementary material, figure S1), while the respective LMs estimate that CP/BP's effect is no more than modest. Likewise, other biotic and intrinsic covariates including moult intensity, age and sex are excluded from consideration by most RFMs (electronic supplementary material, figure S1) and are estimated to have no more than modest effect when included in LMs. Taken together, our findings suggest that reproductively flexible species (e.g. crossbills) can invest in breeding and survival-related processes, which may relate to their ability to exploit abundant food resources.

Ethics. All bird capture and handling protocols were approved by the University of California Davis Institutional Animal Care and Use Committee (protocol number: 16729), US Federal Bird Banding Permit (22712), Wyoming Game & Fish Department (393) and Grand Teton National Park (GRTE-2010-SCI-0004,GRTE-2011-0005SCI, GRTE-2012-SCI-0004, GRTE-2013-SCI-0006).

Data accessibility. Data and R code are available from the Dryad Digital Repository: https://doi.org/10.5061/dryad.v6wwpzgsf [77].

Authors' contributions. E.M.S. performed the immune assays, fieldwork and wrote the manuscript; C.E.G. and E.M.S. carried out the statistical analyses, prepared the figures and tables and edited the manuscript; J.M.C. and D.G.R. participated in fieldwork and contributed to the editing of the manuscript; K.C.K. provided resources and laboratory space for immune assays and participated in editing of the manuscript; T.P.H. provided partial funding, contributed fieldwork and participated in editing of the manuscript. All authors approved the manuscript.

Competing interests. The authors declare no competing interests.

Funding. Many thanks to the funding and support from the University of Wyoming and National Park Service; NSF grant no. 0744705 to T.P.H., and NSF Graduate Research Fellowship; grants from Sigma Xi; Society of Integrative and Comparative Biology; and the American Ornithologists' Union to E.M.S.

Acknowledgements. We thank S.E. Knox, D.Z. Jaul, R.E. Koch, C. Lopez, M. Lohuis and V. Iseri for their help with data collection. We also thank the UW-NPS Research Station and the Murie Center for providing housing and support during field collection.

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
