## [Reviewer comments · Proceedings of the Royal Society B: Biological Sciences]

Review History

RSPB-2018-2450.R0 (Original submission)

Review form: Reviewer 1

Recommendation

Reject – article is not of sufficient interest (we will consider a transfer to another journal)

Is the manuscript an original and important contribution to its field?

Yes

Is the paper of sufficient general interest?

Yes

Is the overall quality of the paper suitable?

No

For papers with colour figures - is colour essential?

Yes

Should the paper be seen by a specialist statistical reviewer?

Yes

Have you any concerns about statistical analyses in this paper? If so, please specify them explicitly in your report. A statement of good statistical practice is available .

Yes

It is a condition of publication that authors make their supporting data, code and materials available. Is it clear from reading the paper how to access all of these?

Yes

Do you have any ethical concerns with this paper?

No

Comments to the Author

The manuscript by Schultz et al. describes patterns of constitutive innate immune function among 4 years and among different seasons within one year in Crossbills, a species that has an extraordinary flexibility in their reproductive timing. The authors have apparently invested a lot of time in catching birds and collected a comprehensive dataset. The merit of this study, which distinguishes it from previous work is that the authors also included data on the main food (cone crop) and on weather variables into their analyses. I definitely appreciate this work and the conceptual idea but have a few comments/concerns (in order of importance).

1) The authors used a two-tiered approach for modelling and report results from RFM, overall LMs and test statistics for each parameter within an LM. Then they report what has and has not been included in each model and what was significant and was not. Very often one reads that certain variables had been identified as having high importance in the RFMs, but then eventually most paragraphs within the result section end with phrases like “these variables were not significant predictors in the subsequent LMs”. And indeed if I look at the supplementary tables, then for the analyses among years, only year is significant for all 4 immune parameters but none of the environmental variables. For the within-year (among season) analyses, none of the predictors is significant for lysis, agglutination and WBC. Only for hapotlobin, precipitation was significant. Given those results, I wonder how the authors can conclude that variation in immune function is “most sensitive to environmental fluctuations – specially food resources, precipitation and temperature” (e.g. line 29-30). To me this looks like yes, there is variation between years and that coincides with cone crop but the conclusion that weather has a significant influence seems not correct given the LMs. Finally, the many different types of models give the impression of fishing for some correlated parameters.

2) In a high cone crop year there is also lots of breeding, in low cone years is very little breeding. Hence, strictly speaking we cannot tease apart which causes the changes in immune function. It could be either the higher food supply or the reproduction itself, even if the measures of reproduction the authors took did not correlate with immune function because their measure is only qualitative but does not allow any assessment of amount of parental investment (as the authors discuss themselves).

3) Is cone production related to weather variables? Looking at Figure 1 it seems like there might be more (spring) precipitation in a good cone year and fewest precipitation in spring 2012 when the worst cone year followed. If so, then it is difficult to disentangle the effect of weather from food supply.

4) No birds were caught multiple times, hence we don't know if the variation is due to individual adjustments/adaptations or due to different parts of the (meta)population being present (or catchable). For example, the authors acknowledge that red crossbill abundance fluctuates between years (line 121). Furthermore different vocal types were caught (it remains unclear which vocal types were caught in which year and/or season). This may mean that apparent

changes in immune function might be driven by different parts of the population being sampled. Alternatively (and not mutually exclusive), different densities may produce different pressure on the immune system which then would coincide with food supply.

5) The authors did elaborate statistics with respect to the immune parameters, but only describe the variation in cone crop. Why not presenting statistical test that confirm which is a large cone crop year and which are low crop years?

6) In general I found the discussion too often just describing the results and the discussion very close to the own results. A broader integration and conceptual perspective would benefit the discussion.

7) The table and Figure captions are not self-explanatory. Please improve them by e.g. mention species name. For Fig 2 it should include that the different seasons have only been measured in one year. For table 1, please make explicit that the data for the seasons are only from one year, etc.

Specific comments

- Line 29: “demonstrate”; you might want to tone down as your data are correlative and you have not done experiments; suggest might be more appropriate
- Line 31: just reading the abstract it is not clear what those physiological factors are. Most people might think of hormones, oxidative stress, telomeres or other physiological systems.
- Line 38: change to “most organisms”; species around the equator or in deep oceans don’t experience extensive seasonal variation.
- Line 60: I think you mean annual-cycle stage here and not life-cycle stage
- Line 86 harsh winter; I suggest to change this to cold winter (to match mild summers); if food is so abundant that birds can breed, the conditions can’t really be harsh
- Line 88: it might be good to briefly mention here already what those environmental and physiological factors are
- Line 112: please add some info on whether one birds makes multiple broods or if the breeding season is long because different individuals breed at different times
- Line 118 and 181: please write Wyoming instead of WY (unless you only expect readers from within the US ;))
- Line 132: here you write that you took 300 ul of blood but later you report that you had to reduce the plasma used for assays because of plasma shortage. Does it mean you used plasma for other studies? No problem with that and I generally appreciate if samples are used for multiple studies/purposes but it might be worth to report that to solve this discrepancy.
- Line 162-165: consider moving this part to after line 129. It doesn’t really fit under the heading “environmental measures”
- Line 178-179: If you only accessed those weather variable for “each day of bird capture”, how could you analyse 1,2,4,8,16 and 32 rolling windows? I guess you mean for bird capture day and those days before.
- Line 183-184: what is the rationale for taking the temperature difference?
- Line 206: I am impressed by your low inter-plate variation. Well done!
- Line 207: what is the percentage of zero scores and what is the spread in the non-zero scores?
- Line 225: immune parameters or measures (but not responses); responses relate to immune challenges
- Line 230: you have not mentioned yet that you did measure haematocrit
- Line 238 and 247: what do you mean with response? Immune parameter?
- Lines 311-318: this paragraphs reads like a results section. Consider moving it to the results.
- Line 340: this info that the within-year analyses (2011) was done in a large cone crop year might better come earlier (in intro or methods already).
- Line 375: “correlate with”, you can not prove causal relationships;
- Line 400-401: this seems trivial: if the cost of winter reproduction would not be manageable, then the birds would not breed in winter!

- Figure 1B: if I understood the data analyses correctly, then two datasets have been used. One for all 4 summers and one for all of 2011. Hence I suggest to remove the winter 2012 data from the figure as those are not part of the analyses. Or do they belong to one winter 2011/12? If they have been treated as one winter in the analyses, then the data should be pooled in the figure as well and not presented separately.
- Figure 1D: why do you show data from 2014?
- Figure S3: the axis is in parts unreadable
- Figure S4: please explain the abbreviations in the figure caption/legend.

Review form: Reviewer 2

Recommendation

Major revision is needed (please make suggestions in comments)

Is the manuscript an original and important contribution to its field?

Yes

Is the paper of sufficient general interest?

Yes

Is the overall quality of the paper suitable?

Yes

Have you any concerns about statistical analyses in this paper? If so, please specify them explicitly in your report. A statement of good statistical practice is available .

No

It is a condition of publication that authors make their supporting data, code and materials available. Is it clear from reading the paper how to access all of these?

Yes

Do you have any ethical concerns with this paper?

No

Comments to the Author

The study presented in manuscript RSPB-2018-2450 uses a species with an uncommon natural history to test alternative hypotheses from ecoimmunology about the proximate causes of investment in immune defenses; a beautiful example of the Krogh principle. Specifically, the study tests whether life cycle events (i.e., reproduction, moulting) or environmental traits have a stronger influence on investment in a variety of immune defenses. Disentangling these hypotheses is typically difficult because most species time their annual cycle to correspond with changes in the environment and the effects of the environment and annual cycle events are conflated. This study, however, utilizes red crossbills, which can breed during the summer and winter because they facultatively respond to the abundance of cone crops, their major food source. In short, this manuscript has the potential to advance our understanding of the proximate effects that drive investment in immunology.

The paper is generally clearly written and I appreciate that it includes a section about the limitations of the sampling regime used (lines 388-434). I have a few major suggestions followed by minor comments.

1) The interpretation of the results rests on the assumption that life cycle can be disentangled from environmental traits because red crossbills can breed during both the summer and winter. The manuscript included a matrix of the correlations between the environmental and life cycle

traits used in this study, but no evidence is provided that measure of reproduction, moult, or physiological condition do not systematically vary with year or season. It could be that associations between covariates and immune responses are not shown statistically because the main categorical trait (i.e., season or year) is masking the relationship and hiding colinearity between traits. To strengthen the argument that this study disentangles the effects of physiology, life cycle stage, and environment, the lack of an association between time (season and year) and the covariates needs to be shown.

2) The discussion is well rooted in the literature about red crossbills. Much research has been conducted on immune defenses in wild animals and in captive, non-model animals, yet little of this research is discussed in the discussion section. I recommend using these studies as context in interpreting the results. For example, the cost of moult are discussed in lines 410-415. Many studies have demonstrated the costs of moult (e.g., Epting 1980 *Physiol Zool*, Murphy 1996 in *Avian energetics and nutritional ecology*) and many have investigated the trade-off between moult and immune defenses (e.g., Moreno et al. 2001 *Oecologia*, Pap et al. 2009 *J Exp Biol*, Ben-Hamo et al. 2017 *J Avian Biol*). These results would provide insights into why it is important to try to capture animals during times of "peak" energy demands. Similarly, published studies might help in interpreting the results about PIT54 (lines 321-331, 363-374).

3) The description of the methods for the white blood cell counts indicate that differential counts were performed, yet data for the differential counts are not presented. Including these data might provide additional insights into patterns for the other immune assays given that different types of white blood cells perform different functions and can change differently in response to traits measured (e.g., Matson et al. 2006 *Proceedings B*).

4) One of the main conclusions is that the variation in immune defenses seen across years matches with variation seen in cone crop. Cone crop was included as a covariate in the models, but was never a significant predictor of the immune defenses (as far as I can tell), potentially because cone crop is conflated with season and year (see critique 1 above). Similarly, although data about cone index counts are presented, differences across seasons and years were never tested statistically. Please provide these statistics to strengthen the argument presented in lines 311-318.

Alternatively, you can perform regression between cone count and the year mean of each immune response to test the relationship presented in lines 311-318.

5) In figures 1B, it appears that there is only complete data for 2011, but summer data are available for all 4 years of the study. Clarify the sample sizes per season per year by extending table 1 and discuss how these missing data might limit the interpretation of the results. The imbalance in the design might be hiding important patterns. Similarly, are all of the seasonal effects being driven by 2011?

Minor comments:

Lines 131-135: Partition this into more than one sentence to improve clarity.

Line 206-208: Thank you for including inter-plate variations.

Lines 325-326: Move the parenthetical so it is after "immune parameters".

Lines 324-327: Are there other studies that support this idea? Arsnoe et al. 2011 *PLoS One* and Downs et al. 2015 *PLoS One* come to mind.

Lines 343-347: A complex, confusing sentence as written.

Lines 366-369: Provide a reference for this idea.

Lines 376-380: What is the number of the reference for Hegemann et al. 2012?

Figure 1B: I think that this figure could be presented in grey-scale if the center symbol (currently a dot) of the cone and bird surveys was changed to a different symbol for one of the items.

Supplemental materials: Provide definitions of abbreviations for abbreviations used in each table or figure. The lack of definitions makes it very difficult to interpret the results.

Supplemental tables: inconsistent bolding of significant results.

Figure S1: Provide the duration of the window of the rolling means.

Figure S2: Fix the overlapping numbers on the x-axis.

I couldn't open the file containing the R code and data because my computer didn't recognize the .7z extension.

Decision letter (RSPB-2018-2450.R0)

28-Nov-2018

Dear Dr Schultz:

I am writing to inform you that your manuscript RSPB-2018-2450 entitled "Patterns of annual and seasonal immune investment in a temporal reproductive opportunist" has, in its current form, been rejected for publication in Proceedings B.

This action has been taken on the advice of referees, who have recommended that very substantial revisions are necessary. With this in mind we would be happy to consider a resubmission, provided the comments of the referees are fully addressed. However please note that this is not a provisional acceptance.

Sincerely,
Professor Hans Heesterbeek
Editor, Proceedings B
<mailto:proceedingsb@royalsociety.org>

Associate Editor, Dr Susannah French
Board Member: 1
Comments to Author:

This manuscript has been reviewed by myself and two experts in the field of ecoimmunology and avian physiology. Myself and both reviewers were impressed with the scale of the data set, and inclusion of environmental factors along with immune measures. However, as you can see from the reviews, both reviewers raised significant concerns with the study design and interpretation of the results based on the data set. The main concerns raised by the reviewers are that 1) the design cannot disentangle breeding versus food effects on immunity; 2) statistical analyses to test differences between cone crop seasons or years were not present, or included as a significant predictor of immunity?; 3) data appear to only be complete for 2011 and no other years (which only appear to have summer data); 4) The period of summer sampling is especially long relative to the fall, is there is there temporal environmental variation during that prolonged window?;

and 5) the discussion should be broadened to include literature and species beyond the current study. In addition, there were other important issues raised by the reviewers that also should be addressed.

Reviewer(s)' Comments to Author:

Referee: 1

Comments to the Author(s)

The manuscript by Schultz et al. describes patterns of constitutive innate immune function among 4 years and among different seasons within one year in Crossbills, a species that has an extraordinary flexibility in their reproductive timing. The authors have apparently invested a lot of time in catching birds and collected a comprehensive dataset. The merit of this study, which distinguishes it from previous work is that the authors also included data on the main food (cone crop) and on weather variables into their analyses. I definitely appreciate this work and the conceptual idea but have a few comments/concerns (in order of importance).

1) The authors used a two-tiered approach for modelling and report results from RFM, overall LMs and test statistics for each parameter within an LM. Then they report what has and has not been included in each model and what was significant and was not. Very often one reads that certain variables had been identified as having high importance in the RFMs, but then eventually most paragraphs within the result section end with phrases like "these variables were not significant predictors in the subsequent LMs". And indeed if I look at the supplementary tables, then for the analyses among years, only year is significant for all 4 immune parameters but none of the environmental variables. For the within-year (among season) analyses, none of the predictors is significant for lysis, agglutination and WBC. Only for hapotlobin, precipitation was significant. Given those results, I wonder how the authors can conclude that variation in immune function is "most sensitive to environmental fluctuations – specially food resources, precipitation and temperature" (e.g. line 29-30). To me this looks like yes, there is variation between years and that coincides with cone crop but the conclusion that weather has a significant influence seems not correct given the LMs. Finally, the many different types of models give the impression of fishing for some correlated parameters.

2) In a high cone crop year there is also lots of breeding, in low cone years is very little breeding. Hence, strictly speaking we cannot tease apart which causes the changes in immune function. It could be either the higher food supply or the reproduction itself, even if the measures of reproduction the authors took did not correlate with immune function because their measure is only qualitative but does not allow any assessment of amount of parental investment (as the authors discuss themselves).

3) Is cone production related to weather variables? Looking at Figure 1 it seems like there might be more (spring) precipitation in a good cone year and fewest precipitation in spring 2012 when the worst cone year followed. If so, then it is difficult to disentangle the effect of weather from food supply.

4) No birds were caught multiple times, hence we don't know if the variation is due to individual adjustments/adaptations or due to different parts of the (meta)population being present (or catchable). For example, the authors acknowledge that red crossbill abundance fluctuates between years (line 121). Furthermore different vocal types were caught (it remains unclear which vocal types were caught in which year and/or season). This may mean that apparent changes in immune function might be driven by different parts of the population being sampled. Alternatively (and not mutually exclusive), different densities may produce different pressure on the immune system which then would coincide with food supply.

5) The authors did elaborate statistics with respect to the immune parameters, but only describe

the variation in cone crop. Why not presenting statistical test that confirm which is a large cone crop year and which are low crop years?

6) In general I found the discussion too often just describing the results and the discussion very close to the own results. A broader integration and conceptual perspective would benefit the discussion.

7) The table and Figure captions are not self-explanatory. Please improve them by e.g. mention species name. For Fig 2 it should include that the different seasons have only been measured in one year. For table 1, please make explicit that the data for the seasons are only from one year, etc.

Specific comments

- Line 29: “demonstrate”; you might want to tone down as your data are correlative and you have not done experiments; suggest might be more appropriate
- Line 31: just reading the abstract it is not clear what those physiological factors are. Most people might think of hormones, oxidative stress, telomeres or other physiological systems.
- Line 38: change to “most organisms”; species around the equator or in deep oceans don’t experience extensive seasonal variation.
- Line 60: I think you mean annual-cycle stage here and not life-cycle stage
- Line 86 harsh winter; I suggest to change this to cold winter (to match mild summers); if food is so abundant that birds can breed, the conditions can’t really be harsh
- Line 88: it might be good to briefly mention here already what those environmental and physiological factors are
- Line 112: please add some info on whether one birds makes multiple broods or if the breeding season is long because different individuals breed at different times
- Line 118 and 181: please write Wyoming instead of WY (unless you only expect readers from within the US ;)
- Line 132: here you write that you took 300 ul of blood but later you report that you had to reduce the plasma used for assays because of plasma shortage. Does it mean you used plasma for other studies? No problem with that and I generally appreciate if samples are used for multiple studies/purposes but it might be worth to report that to solve this discrepancy.
- Line 162-165: consider moving this part to after line 129. It doesn’t really fit under the heading “environmental measures”
- Line 178-179: If you only accessed those weather variable for “each day of bird capture”, how could you analyse 1,2,4,8,16 and 32 rolling windows? I guess you mean for bird capture day and those days before.
- Line 183-184: what is the rational for taking the temperature difference?
- Line 206: I am impressed by your low inter-plate variation. Well done!
- Line 207: what is the percentage of zero scores and what is the spread in the non-zero scores?
- Line 225: immune parameters or measures (but not responses); responses relate to immune challenges
- Line 230: you have not mentioned yet that you did measure haematocrit
- Line 238 and 247: what do you mean with response? Immune parameter?
- Lines 311-318: this paragraphs reads like a results section. Consider moving it to the results.
- Line 340: this info that the within-year analyses (2011) was done in a large cone crop year might better come earlier (in intro or methods already).
- Line 375: “correlate with”, you can not prove causal relationships;
- Line 400-401: this seems trivial: if the cost of winter reproduction would not be manageable, then the birds would not breed in winter!
- Figure 1B: if I understood the data analyses correctly, then two datasets have been used. One for all 4 summers and one for all of 2011. Hence I suggest to remove the winter 2012 data from the figure as those are not part of the analyses. Or do they belong to one winter 2011/12? If they have been treated as one winter in the analyses, then the data should be pooled in the figure as well and not presented separately.
- Figure 1D: why do you show data from 2014?

- Figure S3: the axis is in parts unreadable
- Figure S4: please explain the abbreviations in the figure caption/legend.

Referee: 2

Comments to the Author(s)

The study presented in manuscript RSPB-2018-2450 uses a species with an uncommon natural history to test alternative hypotheses from ecoimmunology about the proximate causes of investment in immune defenses; a beautiful example of the Krogh principle. Specifically, the study tests whether life cycle events (i.e., reproduction, moulting) or environmental traits have a stronger influence on investment in a variety of immune defenses. Disentangling these hypotheses is typically difficult because most species time their annual cycle to correspond with changes in the environment and the effects of the environment and annual cycle events are conflated. This study, however, utilizes red crossbills, which can breed during the summer and winter because they facultatively respond to the abundance of cone crops, their major food source. In short, this manuscript has the potential to advance our understanding of the proximate effects that drive investment in immunology.

The paper is generally clearly written and I appreciate that it includes a section about the limitations of the sampling regime used (lines 388-434). I have a few major suggestions followed by minor comments.

- 1) The interpretation of the results rests on the assumption that life cycle can be disentangled from environmental traits because red crossbills can breed during both the summer and winter. The manuscript included a matrix of the correlations between the environmental and life cycle traits used in this study, but no evidence is provided that measure of reproduction, moult, or physiological condition do not systematically vary with year or season. It could be that associations between covariates and immune responses are not shown statistically because the main categorical trait (i.e., season or year) is masking the relationship and hiding colinearity between traits. To strengthen the argument that this study disentangles the effects of physiology, life cycle stage, and environment, the lack of an association between time (season and year) and the covariates needs to be shown.
- 2) The discussion is well rooted in the literature about red crossbills. Much research has been conducted on immune defenses in wild animals and in captive, non-model animals, yet little of this research is discussed in the discussion section. I recommend using these studies as context in interpreting the results. For example, the cost of moult are discussed in lines 410-415. Many studies have demonstrated the costs of moult (e.g., Epting 1980 *Physiol Zool*, Murphy 1996 in *Avian energetics and nutritional ecology*) and many have investigated the trade-off between moult and immune defenses (e.g., Moreno et al. 2001 *Oecologia*, Pap et al. 2009 *J Exp Biol*, Ben-Hamo et al. 2017 *J Avian Biol*). These results would provide insights into why it is important to try to capture animals during times of "peak" energy demands. Similarly, published studies might help in interpreting the results about PIT54 (lines 321-331, 363-374).
- 3) The description of the methods for the white blood cell counts indicate that differential counts were performed, yet data for the differential counts are not presented. Including these data might provide additional insights into patterns for the other immune assays given that different types of white blood cells perform different functions and can change differently in response to traits measured (e.g., Matson et al. 2006 *Proceedings B*).
- 4) One of the main conclusions is that the variation in immune defenses seen across years matches with variation seen in cone crop. Cone crop was included as a covariate in the models, but was never a significant predictor of the immune defenses (as far as I can tell), potentially because cone crop is conflated with season and year (see critique 1 above). Similarly, although data about cone index counts are presented, differences across seasons and years were never tested statistically. Please provide these statistics to strengthen the argument presented in lines 311-318.

Alternatively, you can perform regression between cone count and the year mean of each immune response to test the relationship presented in lines 311-318.

5) In figures 1B, it appears that there is only complete data for 2011, but summer data are available for all 4 years of the study. Clarify the sample sizes per season per year by extending table 1 and discuss how these missing data might limit the interpretation of the results. The imbalance in the design might be hiding important patterns. Similarly, are all of the seasonal effects being driven by 2011?

Minor comments:

Lines 131-135: Partition this into more than one sentence to improve clarity.

Line 206-208: Thank you for including inter-plate variations.

Lines 325-326: Move the parenthetical so it is after "immune parameters".

Lines 324-327: Are there other studies that support this idea? Arsnoe et al. 2011 PLoS One and Downs et al. 2015 PLoS One come to mind.

Lines 343-347: A complex, confusing sentence as written.

Lines 366-369: Provide a reference for this idea.

Lines 376-380: What is the number of the reference for Hegemann et al. 2012?

Figure 1B: I think that this figure could be presented in grey-scale if the center symbol (currently a dot) of the cone and bird surveys was changed to a different symbol for one of the items.

Supplemental materials: Provide definitions of abbreviations for abbreviations used in each table or figure. The lack of definitions makes it very difficult to interpret the results.

Supplemental tables: inconsistent bolding of significant results.

Figure S1: Provide the duration of the window of the rolling means.

Figure S2: Fix the overlapping numbers on the x-axis.

I couldn't open the file containing the R code and data because my computer didn't recognize the .7z extension.

Author's Response to Decision Letter for (RSPB-2018-2450.R0)

See Appendix A.

RSPB-2019-1238.R0

Review form: Reviewer 3

Recommendation

Major revision is needed (please make suggestions in comments)

Scientific importance: Is the manuscript an original and important contribution to its field?

Acceptable

General interest: Is the paper of sufficient general interest?

Acceptable

Quality of the paper: Is the overall quality of the paper suitable?

Marginal

Is the length of the paper justified?

Yes

Should the paper be seen by a specialist statistical reviewer?

No

Do you have any concerns about statistical analyses in this paper? If so, please specify them explicitly in your report.

Yes

It is a condition of publication that authors make their supporting data, code and materials available - either as supplementary material or hosted in an external repository. Please rate, if applicable, the supporting data on the following criteria.

Is it accessible?

Yes

Is it clear?

Yes

Is it adequate?

Yes

Do you have any ethical concerns with this paper?

No

Comments to the Author

GENERAL COMMENTS TO THE AUTHORS:

The study ("Patterns of annual and seasonal immune investment in a temporal reproductive opportunist" explores variation of red crossbills (*Loxia curvirostra*) immune parameters in response to individual and environmental conditions based on bird individuals captured during four years (2010-2013). Birds were mostly captured in summer, but captured in different seasons in one of the study 'cone years' (2011). I agree with the other reviewers that this study uses an interesting model system to explore the timely question of whether immune investment varies in response to environmental conditions, resource availability and individual condition/reproduction. While I also appreciate the considerable efforts the authors have undertaken, I was unfortunately not able to understand all details of the analysis (which I found somewhat inconsistent) and was not convinced by the results. This is because of the following main reasons:

- 1) The authors used basic Spearman correlation tests to explore any relationships between their immune response variables and climate conditions, according to Table S5, all correlation coefficients were ≤ 0.3 , suggesting weak relationships.
- 2) The authors combine random forest models (RFM) with generalized linear models (GLM) for analyzing the relationships between their immune response variables and suites of covariates, including environmental attributes and measures of individuals attributes of captured birds. The intent to use the RFMs for variable selection and the GLMs for inference based on selected covariates. In terms of the modelling procedure, I found it difficult to understand if this approach was actually necessary but more importantly, I was lost in the results section: some of the statements in the results are based on the relative importance of variables in the RFMs but it is not clear if these are meaningful results at all: if all variables would have little explanatory power, even those with the highest relative importance would be meaningless in explaining variation in the response variables? What I missed in the results were clear statements that covariates had significant effects for explaining variation in the response variables, based on coefficient estimates being clearly distinct from zero or model-based inference (based on information criteria, for GLM, for example).
- 3) The authors state that 'cone year' had most impact on variation in their immune measures, which indicates some variation over years. I agree with reviewer 1 that such relationships does not tell anything about immune investment in response to resource availability unless I missed

some relevant results? I therefore was not able to find strong support for any conclusions on the role of food resources as stated in line 35 (abstract).

Please apologize if I misinterpreted your approaches and results, in which case I am looking forward to learn from the authors' responses.

Please find specific comments below (most comments are focused on the methods as I think the results section need a major revision to report clear coefficient effects). Hopefully, some suggestions are useful to the authors.

SPECIFIC COMMENTS:

Lines 30-31 You mention "immune investment" and "immune variation" without going into more details which particular expressions etc have been investigated, and I found this a bit vague.

Line 34 Perhaps consider to mention the particular component of immunity found to Exhibit seasonal variation? I found this too vague without reading anything in the abstract about the immunity components explored in this study.

Line 36 Please check the expression "physiological measures such as reproduction": perhaps 'reproduction' would be better described as a demographic measure?

Line 48 Perhaps replace "disease potential" with "disease exposure" or "disease susceptibility"?

Line 68 The term "stages of the annual cycle" is not entirely clear - 'stages' of what?

Lines 117-119 I don't think that the statement "RFMs, however, lack the familiarity and robust inference framework of LMs. As such, we used the variables selected by each RFM to build a corresponding LM" is necessarily true. Can you provide any representative reference for this? If LM are more robust for inference (in your opinion), why don't you practice variable selection in an LM framework rather than RFM? Suggest to delete this statement, which is also of limited relevance in the introduction.

Line 132 Which "nine month"?

Lines 221-231 I do not fully understand how your Spearman correlations test were linked to the rest of the analysis (RFM and LMs)? Did you aim to select time windows for averaging weather variables based on strongest correlations? Overall, I think this approach needs some clarification. Moreover, I think it would help to mention all statistical analysis in the paragraph "Statistical Analysis". Perhaps it will then also become clear why you do all the separate correlation test while also using random forest models as a variable selection tool.

Line 222 Which window length(s) were considered 'plausible' in your particular study and for your study organisms? Please specify.

Line 265 Provide the version number of R used in your study.

Line 265 Which thresholds was used for "where not highly correlated"? Suggest to state this here.

Line 277 Confusing to mention "cone year" in context of analysing samples from summer periods, which do not match "cone years"?

Line 283 Please check: while you refer to Table S3 as an overview of "all measured covariates" the legend of Table S3 states "List of covariates selected by random forest models".

Lines 299-307 I do not understand your arguments here and I am confused: if random forest are conducted as a tool for variable selection, why are you discussing variable/model selection for GLMs? Also, there is certainly also literature that emphasise the problems that collinearity can cause in terms of coefficient estimates. Again, why not avoiding any collinearity issues given you multi-step approach?

Given the relatively small number of covariates, you may also state which variables were strongly correlated.

Lines 306 309 What do you mean by "time period"? This term has not been defined before and is unclear.

Lines 310 Please check: should "LM" be replaced by "GLM" (given that you also use logistic regression models)?

Legend Table S2 I found the description "Table S2: Summary of observations..." confusing: are you talking about the number of captures or the number of captured individuals. If this table list the number of captures, how did you deal with recaptures in your analysis?

Lines 311 We did you choose " $p < 0.2$ "? This value seems to be high given that many studies use a threshold of $p = 0.5$ or $p = 0.01$ for concluding about 'significance'. I think your value

warrants some justification. Also, I suggest to mention the test statistics on which these p-values are based.

Decision letter (RSPB-2019-1238.R0)

I am writing to inform you that this version of your manuscript RSPB-2019-1238 entitled "Patterns of annual and seasonal immune investment in a temporal reproductive opportunist" has, in its current form, been rejected for publication in Proceedings B.

This action has been taken on the advice of a referee and the Associate Editor, who have recommended that substantial revisions are necessary. With this in mind we would be happy to consider a resubmission, provided the comments of the referees are fully addressed. However please note that this is not a provisional acceptance. Please also note that I have made an exception by granting you a second major revision. The criticism by the referee and the Associate Editor are such that they would in most cases with comparable opinions after a round of major revision lead to rejection of the manuscript. Because some of the issues revolve around clarification of the statistics, we have decided to give you the opportunity to respond and revise. We may seek the advice of a statistical reviewer if this is necessary to reach a decision in the next round.

Please find below the comments made by the referees, not including confidential reports to the Editor, which I hope you will find useful.

Sincerely,
 Professor Hans Heesterbeek
 mailto: proceedingsb@royalsociety.org

Associate Editor
 Comments to Author:

The authors have done a lot of work to address prior concerns by reviewers and have helped clarify many points and have greatly improved the quality and clarity of the manuscript. However, there are still significant clarifications and concerns with the analytical approaches. The scope of the field sampling is greater than what most current researchers in this field have

done, which makes this paper significant. However, the authors are finding little explanatory value in the variables they are investigating. More specifically the most significant issues remaining are 1) the combined use of GLM and RFM still needs further clarification as both myself and the reviewer were unclear as to the approach and interpretations based in these analyses; and 2) I agree with the overreliance on strict 0.05 p values in the field today, however many of the relationships found in the current study were weak (>0.2) and thus some of interpretations seem overly strong based on the statistical findings. In addition, there is also a list of more specific issues raised by the reviewer.

Reviewer(s)' Comments to Author:

Referee: 3

Comments to the Author(s).

GENERAL COMMENTS TO THE AUTHORS:

The study ("Patterns of annual and seasonal immune investment in a temporal reproductive opportunist" explores variation of red crossbills (*Loxia curvirostra*) immune parameters in response to individual and environmental conditions based on bird individuals captured during four years (2010-2013). Birds were mostly captured in summer, but captured in different seasons in one of the study 'cone years' (2011). I agree with the other reviewers that this study uses an interesting model system to explore the timely question of whether immune investment varies in response to environmental conditions, resource availability and individual condition/reproduction. While I also appreciate the considerable efforts the authors have undertaken, I was unfortunately not able to understand all details of the analysis (which I found somewhat inconsistent) and was not convinced by the results. This is because of the following main reasons:

- 1) The authors used basic Spearman correlation tests to explore any relationships between their immune response variables and climate conditions, according to Table S5, all correlation coefficients were < 0.3 , suggesting weak relationships.
- 2) The authors combine random forest models (RFM) with generalized linear models (GLM) for analyzing the relationships between their immune response variables and suites of covariates, including environmental attributes and measures of individuals attributes of captured birds. The intent to use the RFMs for variable selection and the GLMs for inference based on selected covariates. In terms of the modelling procedure, I found it difficult to understand if this approach was actually necessary but more importantly, I was lost in the results section: some of the statements in the results are based on the relative importance of variables in the RFMs but it is not clear if these are meaningful results at all: if all variables would have little explanatory power, even those with the highest relative importance would be meaningless in explaining variation in the response variables? What I missed in the results were clear statements that covariates had significant effects for explaining variation in the response variables, based on coefficient estimates being clearly distinct from zero or model-based inference (based on information criteria, for GLM, for example).
- 3) The authors state that 'cone year' had most impact on variation in their immune measures, which indicates some variation over years. I agree with reviewer 1 that such relationships does not tell anything about immune investment in response to resource availability unless I missed some relevant results? I therefore was not able to find strong support for any conclusions on the role of food resources as stated in line 35 (abstract).

Please apologize if I misinterpreted your approaches and results, in which case I am looking forward to learn from the authors' responses.

Please find specific comments below (most comments are focused on the methods as I think the results section need a major revision to report clear coefficient effects). Hopefully, some suggestions are useful to the authors.

SPECIFIC COMMENTS:

Lines 30-31 You mention “immune investment” and “immune variation” without going into more details which particular expressions etc have been investigated, and I found this a bit vague.

Line 34 Perhaps consider to mention the particular component of immunity found to Exhibit seasonal variation? I found this too vague without reading anything in the abstract about the immunity components explored in this study.

Line 36 Please check the expression “physiological measures such as reproduction”: perhaps ‘reproduction’ would be better described as a demographic measure?

Line 48 Perhaps replace “disease potential” with “disease exposure” or “disease susceptibility”?

Line 68 The term “stages of the annual cycle” is not entirely clear – ‘stages’ of what?

Lines 117-119 I don’t think that the statement “RFMs, however, lack the familiarity and robust inference framework of LMs. As such, we used the variables selected by each RFM to build a corresponding LM” is necessarily true. Can you provide any representative reference for this? If LM are more robust for inference (in your opinion), why don’t you practice variable selection in an LM framework rather than RFM? Suggest to delete this statement, which is also of limited relevance in the introduction.

Line 132 Which “nine month”?

Lines 221-231 I do not fully understand how your Spearman correlations test were linked to the rest of the analysis (RFM and LMs)? Did you aim to select time windows for averaging weather variables based on strongest correlations? Overall, I think this approach needs some clarification. Moreover, I think it would help to mention all statistical analysis in the paragraph “Statistical Analysis”. Perhaps it will then also become clear why you do all the separate correlation test while also using random forest models as a variable selection tool.

Line 222 Which window length(s) were considered ‘plausible’ in your particular study and for your study organisms? Please specify.

Line 265 Provide the version number of R used in your study.

Line 265 Which thresholds was used for “where not highly correlated”? Suggest to state this here.

Line 277 Confusing to mention “cone year” in context of analysing samples from summer periods, which do not match “cone years”?

Line 283 Please check: while you refer to Table S3 as an overview of “all measured covariates” the legend of Table S3 states “List of covariates selected by random forest models”.

Lines 299-307 I do not understand your arguments here and I am confused: if random forest are conducted as a tool for variable selection, why are you discussing variable/model selection for GLMs? Also, there is certainly also literature that emphasise the problems that collinearity can cause in terms of coefficient estimates. Again, why not avoiding any collinearity issues given you multi-step approach?

Given the relatively small number of covariates, you may also state which variables were strongly correlated.

Lines 306 309 What do you mean by “time period”? This term has not been defined before and is unclear.

Lines 310 Please check: should “LM” be replaced by “GLM” (given that you also use logistic regression models)?

Legend Table S2 I found the description “Table S2: Summary of observations...” confusing: are you talking about the number of captures or the number of captured individuals. If this table list the number of captures, how did you deal with recaptures in your analysis?

Lines 311 We did you choose “ $p < 0.2$ ”? This value seems to be high given that many studies use a threshold of $p = 0.5$ or $p = 0.01$ for concluding about ‘significance’. I think your value warrants some justification. Also, I suggest to mention the test statistics on which these p-values are based.

Author's Response to Decision Letter for (RSPB-2019-1238.R0)

See Appendix B.

RSPB-2019-2993.R0

Review form: Reviewer 4

Recommendation

Major revision is needed (please make suggestions in comments)

Scientific importance: Is the manuscript an original and important contribution to its field?

Good

General interest: Is the paper of sufficient general interest?

Good

Quality of the paper: Is the overall quality of the paper suitable?

Good

Is the length of the paper justified?

Yes

Should the paper be seen by a specialist statistical reviewer?

Yes

Do you have any concerns about statistical analyses in this paper? If so, please specify them explicitly in your report.

No

It is a condition of publication that authors make their supporting data, code and materials available - either as supplementary material or hosted in an external repository. Please rate, if applicable, the supporting data on the following criteria.

Is it accessible?

Yes

Is it clear?

Yes

Is it adequate?

Yes

Do you have any ethical concerns with this paper?

No

Comments to the Author

RSPB-2019-2993 ("Predictors of immune variation") by Schultz et al. is a generally clearly written manuscript that characterises within and among year variation in a wild passerine. The analysis is extensive, with much material included as supplementary material. For me, the quantity of analyses and materials presented makes a critical evaluation of the discussion (and results) a bit challenging. (And I've highlighted examples of apparent conflicts below.) Nevertheless, I think this work is a valuable and comprehensive contribution to the field of ecological immunology/physiology.

One of my bigger concerns in relation to the publication of this manuscript in RSPB is the broader context in which the authors place their work. In its current form, it seems to me that several highly relevant references have not been included in the introduction (and discussion), and one

result is that the novelty of the work feels slightly exaggerated. For example, L59-60 states "studies of birds have focussed on single life-history stages in the annual cycle." This is not true: one can look to the lab of Tielemann, which has examined immune function across annual cycles.

E.g.:

Immune function in a free-living bird varies over the annual cycle, but seasonal patterns differ between years

<https://link.springer.com/article/10.1007/s00442-012-2339-3>

(This paper is cited in other contexts, but it contradicts L-59-60 statement. Same is true of reference 63, Buehler et al.)

But, there are other papers too, e.g.:

Genetic and phenotypically flexible components of seasonal variation in immune function

<https://jeb.biologists.org/content/217/9/1510>

Perhaps, even more relevant is the group's work on tropical birds that relates to what Schultz et al. refer to as "disentangling the effects of environment and physiological processes on immune investment. Examples of existing work include the following:

Constitutive innate immunity of tropical House Wrens varies with season and reproductive activity

<https://academic.oup.com/auk/article/136/3/ukz029/5486174>

No downregulation of immune function during breeding in two year-round breeding bird species in an equatorial East African environment

<https://onlinelibrary.wiley.com/doi/full/10.1111/jav.02151>

Seasonal differences in baseline innate immune function are better explained by environment than annual cycle stage in a year-round breeding tropical songbird

<https://besjournals.onlinelibrary.wiley.com/doi/pdf/10.1111/1365-2656.12948>

Geographical and temporal variation in environmental conditions affects nestling growth but not immune function in a year-round breeding equatorial lark

<https://frontiersinzoology.biomedcentral.com/articles/10.1186/s12983-017-0213-1>

Incorporating what is already known from these papers (and possibly others) will add important context to the manuscript.

In the sub-section "Local Weather Conditions" (and L222), I was a bit surprised to read that conditions were only recorded/used in analyses over the 24 hours around capture. It seems unrealistic to me to think that the temperature of the day of capture, and not the week or month (e.g., average) before capture would be important. I would expect possible lag effects here. Can the authors better justify this decision?

I was also a bit puzzled by the decision to "focus on mean yearly cone crop scores," when the four vocal types studied each specialise on a different cone type. This seems to be further complicated by the fact that the vocal types are not equally represented in the study. Type 5 was most commonly sampled, and this type prefers Lodgepole Pine and Engelmann Spruce over Douglas Fir and Blue Spruce <https://ebird.org/news/recrtype/>. Would it not make more sense to focus on the preferred resources?

In the discussion, I find a particular passage (L374-381) quite confusing. How/why do the authors propose that birds would use a higher cost defence when food resources are low? Doesn't this go against all trade-off theory? The authors try to reconcile this by explaining that the costs of the 'lower-cost options' (I'm paraphrasing here) are higher than an "expensive inflammatory response." To me, the reasoning here comes across quite circular and suggests that "high cost"

and “low cost” defences have been incorrectly defined/categorised. The bottom line is this text could/should be clarified.

Minor comments:

L74: Should this be specifically “inter-annual variation”?

L165-167: Can more details (perhaps methodological data) be provided to waylay any concerns of readers that having a second observer of one season (summer 2012) did not lead to any systematic differences?

L171: “Poor predictive value” of what? Can you explain a bit more thoroughly?

L303, L305, L313, L315 (elsewhere?): Perhaps a point of semantics, but I was a bit confused by the use of “predictors of seasonal variation” and “predicting annual variation.” I don’t think you are predicting (seasonal/annual) *variation*, you are predicting the magnitude of immune indices using models that include season/year and other explanatory variables.

L400-401 and L406-407: Authors write temperatures were highest in summer and precipitation was higher in winter. It’s not clear to me which of these colinear variables (season and climate variable) is driving each of these patterns. With temperature, Fig. 2B suggests it is season, not the temp itself. With precipitation, it seems to be the snow, not the season. L300-301 state no significant effects of season on PIT54, but L410-412 state “higher winter season PIT54 in crossbills. I’m confused. It is unclear to me why this discussion is structured the way it is and emphasises the points that it does.

L417-418: Is another possibility not that good condition allowed for PIT54 production?

L437-440: Please see comments re: introduction missing relevant references. References are also missing here.

L450-452: This concluding sentence seems to come from out of nowhere. There is little attention in the manuscript to topics of mobility and exploitation of food patches.

Decision letter (RSPB-2019-2993.R0)

11-Feb-2020

Dear Dr Schultz:

Your manuscript has now been peer reviewed and the reviews have been assessed by an Associate Editor. The reviewer's comments (not including confidential comments to the Editor) are included at the end of this email for your reference. As you will see, the reviewer has raised some concerns with your manuscript and we would like to invite you to revise your manuscript to address them.

You have made substantial changes in response to the comments by a previous reviewer and the manuscript has now been through two major revisions. This is very rare for Proceedings B, as explained in my previous decision letter. Unfortunately, the critical reviewer was not available for a third look at the manuscript and we sought the opinion of a new reviewer. In light of this, it is only fair to allow you to respond. If the new reviewer would have raised fundamental issues then the review would have led to rejection. Although the reviewer has no fundamental problems, from which I conclude that the explanation of the statistical analysis is now sufficiently

clear, he does raise some issues that speak to novelty. I trust you can address these in the introduction and discussion. As this is going to be the last revision before we take a final decision, I urge you to make every effort to fully address all of the comments at this stage.

If deemed necessary by the Associate Editor, your manuscript will be sent back to one or more of the original reviewers for assessment. If the original reviewers are not available we may invite new reviewers. Please note that we cannot guarantee eventual acceptance of your manuscript at this stage.

Research ethics:

Use of animals and field studies:

Please submit a copy of your revised paper within three weeks. If we do not hear from you within this time your manuscript will be rejected. If you are unable to meet this deadline please let us know as soon as possible, as we may be able to grant a short extension.

Best wishes,
Professor Hans Heesterbeek
mailto:proceedingsb@royalsociety.org

Reviewer(s)' Comments to Author:

Referee: 4

Comments to the Author(s).

RSPB-2019-2993 ("Predictors of immune variation") by Schultz et al. is a generally clearly written manuscript that characterises within and among year variation in a wild passerine. The analysis is extensive, with much material included as supplementary material. For me, the quantity of analyses and materials presented makes a critical evaluation of the discussion (and results) a bit challenging. (And I've highlighted examples of apparent conflicts below.) Nevertheless, I think this work is a valuable and comprehensive contribution to the field of ecological immunology/physiology.

One of my bigger concerns in relation to the publication of this manuscript in RSPB is the broader context in which the authors place their work. In its current form, it seems to me that several highly relevant references have not been included in the introduction (and discussion), and one result is that the novelty of the work feels slightly exaggerated. For example, L59-60 states "studies of birds have focussed on single life-history stages in the annual cycle." This is not true: one can look to the lab of Tieleman, which has examined immune function across annual cycles.

E.g.:

Immune function in a free-living bird varies over the annual cycle, but seasonal patterns differ between years

<https://link.springer.com/article/10.1007/s00442-012-2339-3>

(This paper is cited in other contexts, but it contradicts L-59-60 statement. Same is true of reference 63, Buehler et al.)

But, there are other papers too, e.g.:

Genetic and phenotypically flexible components of seasonal variation in immune function
<https://jeb.biologists.org/content/217/9/1510>

Perhaps, even more relevant is the group's work on tropical birds that relates to what Schultz et al. refer to as "disentangling the effects of environment and physiological processes on immune investment. Examples of existing work include the following:

Constitutive innate immunity of tropical House Wrens varies with season and reproductive activity

<https://academic.oup.com/auk/article/136/3/ukz029/5486174>

No downregulation of immune function during breeding in two year-round breeding bird species in an equatorial East African environment

<https://onlinelibrary.wiley.com/doi/full/10.1111/jav.02151>

Seasonal differences in baseline innate immune function are better explained by environment than annual cycle stage in a year-round breeding tropical songbird

<https://besjournals.onlinelibrary.wiley.com/doi/pdf/10.1111/1365-2656.12948>

Geographical and temporal variation in environmental conditions affects nestling growth but not immune function in a year-round breeding equatorial lark

<https://frontiersinzoology.biomedcentral.com/articles/10.1186/s12983-017-0213-1>

Incorporating what is already known from these papers (and possibly others) will add important context to the manuscript.

In the sub-section "Local Weather Conditions" (and L222), I was a bit surprised to read that conditions were only recorded/used in analyses over the 24 hours around capture. It seems unrealistic to me to think that the temperature of the day of capture, and not the week or month (e.g., average) before capture would be important. I would expect possible lag effects here. Can the authors better justify this decision?

I was also a bit puzzled by the decision to "focus on mean yearly cone crop scores," when the four vocal types studied each specialise on a different cone type. This seems to be further complicated by the fact that the vocal types are not equally represented in the study. Type 5 was most commonly sampled, and this type prefers Lodgepole Pine and Engelmann Spruce over Douglas Fir and Blue Spruce <https://ebird.org/news/recrtype/>. Would it not make more sense to focus on the preferred resources?

In the discussion, I find a particular passage (L374-381) quite confusing. How/why do the authors propose that birds would use a higher cost defence when food resources are low? Doesn't this go against all trade-off theory? The authors try to reconcile this by explaining that the costs of the 'lower-cost options' (I'm paraphrasing here) are higher than an "expensive inflammatory response." To me, the reasoning here comes across quite circular and suggests that "high cost" and "low cost" defences have been incorrectly defined/categorised. The bottom line is this text could/should be clarified.

Minor comments:

L74: Should this be specifically "inter-annual variation"?

L165-167: Can more details (perhaps methodological data) be provided to waylay any concerns of

readers that having a second observer of one season (summer 2012) did not lead to any systematic differences?

L171: "Poor predictive value" of what? Can you explain a bit more thoroughly?

L303, L305, L313, L315 (elsewhere?): Perhaps a point of semantics, but I was a bit confused by the use of "predictors of seasonal variation" and "predicting annual variation." I don't think you are predicting (seasonal/annual) *variation*, you are predicting the magnitude of immune indices using models that include season/year and other explanatory variables.

L400-401 and L406-407: Authors write temperatures were highest in summer and precipitation was higher in winter. It's not clear to me which of these colinear variables (season and climate variable) is driving each of these patterns. With temperature, Fig. 2B suggests it is season, not the temp itself. With precipitation, it seems to be the snow, not the season. L300-301 state no significant effects of season on PIT54, but L410-412 state "higher winter season PIT54 in crossbills. I'm confused. It is unclear to me why this discussion is structured the way it is and emphasises the points that it does.

L417-418: Is another possibility not that good condition allowed for PIT54 production?

L437-440: Please see comments re: introduction missing relevant references. References are also missing here.

L450-452: This concluding sentence seems to come from out of nowhere. There is little attention in the manuscript to topics of mobility and exploitation of food patches.

Author's Response to Decision Letter for (RSPB-2019-2993.R0)

See Appendix C.

RSPB-2019-2993.R1 (Revision)

Review form: Reviewer 5

Recommendation

Major revision is needed (please make suggestions in comments)

Scientific importance: Is the manuscript an original and important contribution to its field?

Excellent

General interest: Is the paper of sufficient general interest?

Good

Quality of the paper: Is the overall quality of the paper suitable?

Acceptable

Is the length of the paper justified?

Yes

Should the paper be seen by a specialist statistical reviewer?

No

Do you have any concerns about statistical analyses in this paper? If so, please specify them explicitly in your report.

Yes

It is a condition of publication that authors make their supporting data, code and materials available - either as supplementary material or hosted in an external repository. Please rate, if applicable, the supporting data on the following criteria.

Is it accessible?

Yes

Is it clear?

Yes

Is it adequate?

Yes

Do you have any ethical concerns with this paper?

No

Comments to the Author

Schultz et. al. studied the predictors of immune variation based on a very comprehensive dataset of crossbill immune parameters and a suite of annual cycle traits and environmental covariates across four consecutive summers and between seasons within one year. The authors clearly state their intention to provide a comprehensive descriptive study that highlights the ecological determinants of immune function and this has been done. I particularly like the idea of a two-tiered approach to the data analyses employed by the authors, splitting analyses into an exploratory and a more focused section (see suggestion below about making the second tier more hypothetical). They use the non-parametric random forest models to determine the covariates that merit detailed attention, and linear models to test the effects of selected variables on measured immune parameters. This was aimed at getting to more concise and less parameterised final models, and yet not losing on the benefit of highlighting all possible parameters that may affect immune function. Authors conclude that immune function varies substantially in crossbills among summers in four consecutive years and this they argue is driven largely by food resources (annual cone abundance or cone crop scores). Within the year between seasons, authors found that there was less pronounced variation in immune indices and there was no dominant predictor among the variables they considered.

I find no obvious errors in the writing, but I think that more can be achieved from the data. I believe that the findings will make a significant contribution to the field of ecological immunology. However, I have some concerns (suggestions included):

1. Given the number of predictor variables considered in the analyses and the descriptive approach that the authors have adopted, it is difficult to make a critical assessment of the generality of the manuscript in its current form. I understand the difficulty of making predictions about variation in immune function within a life history context, but this is important for guiding a reader's expectation and can be made under specific assumptions, especially because the authors have measured a lot of factors that can influence immune function.

I recommend that the authors should focus the second tier of their analyses on the opportunity provided by the 'novelty' of their study system i.e. testing the effect of the interaction between reproduction and cone crop scores on immune function. This they stated in lines 26 – 28 (see also line 84-94) but did not focus on: "The red crossbill, *Loxia curvirostra*, is a songbird that can breed

opportunistically if conifer seeds are abundant, on both short, cold, and long, warm days, providing an ideal system to investigate interactions between immunity, reproduction, and environmental fluctuations." A study cited by the authors did something similar with year-round breeding common bulbuls *Pycnonotus barbatus* in a seasonally arid tropical environment (reference 23). This system presents an even better opportunity since authors have measured the resource of interest in the environment. The novelty of the system may not give any advantage if it is not fully utilised.

The effects of cone availability and breeding on immune investment needs to be disentangled in the statistical analyses and shown in the figures and tables, so that the conclusion, "Taken together, our findings suggest that reproductively flexible species (e.g., crossbills) can simultaneously invest in breeding and survival-related processes, which may relate to their ability to exploit abundant food resources" can hold. It needs to be backed up by a clear test of the interaction between breeding status and cone crop scores at the time of measuring immune parameters or some time before if a lag effect is expected (not just the annual average). Cone crop scores and breeding status (CP/BP) and their interaction should be retained in the second tier models predicting immune parameters. In addition, the 'variables warranting further interest' can be included as confounding variables. The first tier analyses should have already eliminated many non-significant variables, allowing the effect of the now fewer 'variables warranting further interest' to be tested and discussed alongside the main focus - disentangling the effects of environment and physiological processes on immune function. This will help authors to condense the discussion and make it more general than it is now. Authors can then elaborate more on the life history implications of seasonal variation in immune function as revealed from their system and the added value of their approach.

2. I cannot find the results of the random forest models anywhere in the manuscript, except a reference Figure S1 in the discussion. Since this was a clear objective of the study, the results should be presented in the result section.

3. Table 1 needs to be split in two (between inter and intra annual variation) and expanded to show the full model results for each immune parameter - there is no way of knowing the variables tested and whether they were significant from that table as it is (one needs to go to the results in the text). The p-value of a model is not very useful for any inference on the effects of predictor variables as far as I know. Also, it is neater to report p-values in two decimal places or as <0.001 at most (see also supplementary information).

4. Figure 1 and 2 are too busy and the captions are not very clear. There is no reference to Fig 1 B anywhere in the text and it is not clear why this is included on the panel. Also, the use of the term response is confusing. I will suggest that authors should outrightly call this immune parameter or something more meaningful. The grey grid lines can be removed from the plots to make the figures clearer. Authors stated in the methods (259-260) that lysis was analysed as a binary outcome (0 or 1) but in Figure 1 and 2 this is presented otherwise. Figure 1 may be moved to supplementary information section - it informs about the method not the results. Figure 2 needs to be split in two (between inter and intra annual variation) and the relationship between immune parameters and cone crop scores needs to be shown in the main manuscript (perhaps something like figure S5).

5. In line 414 - 422, authors highlight a contradictory relationship for haptoglobin concentration: they state that PIT54 was higher in birds in good condition, yet PIT54 was high in the year with smaller cone scores. It is important to highlight why this may arise if higher body condition and cone score reflect quality condition. See <https://besjournals.onlinelibrary.wiley.com/doi/full/10.1111/1365-2656.13152> for ideas on how similar immune parameters as used in this study may relate differently to nutrition and body mass (condition).

Minor comments

Line 25 – what do authors mean by mild environmental conditions?

It is important to highlight whether all immune assays were carried together or whether samples were analysed differently every year as they were collected. Differences between years may arise from sample handling or storage time, so it is important to highlight that this was considered and controlled for. The Coefficient of variation of control samples or standards should also be reported if used.

Line 295 – can authors highlight the implication of this significant effect of capture duration and whether it can potentially affect the interpretation of their results?

The title of the supplementary information is different from that of the main manuscript.

What significance threshold for p-value is used for the study? Is there a threshold for R2?

Authors state that "For clarity, models with $R^2 < 0.05$ are omitted, and estimates with p-value < 0.1 are bolded." Please, check to be sure that this is right.

Good luck with revisions

Decision letter (RSPB-2019-2993.R1)

27-Apr-2020

Dear Dr Schultz:

Your manuscript has now been peer reviewed. The reviewer's comments (not including confidential comments to the Editor) are included at the end of this email for your reference. The original reviewer (4) was not available. This is a pity as I had hoped to get convergence to an acceptable manuscript at this stage. In my view, your response to the previous reviewer's comments is adequate. It is normal procedure, however, that revised manuscripts automatically get sent to reviewers who have recommended 'major revisions'. The reviewer was not available, but did recommend an alternative expert, who was then contacted by the Editorial office. Normally this is fine, but new reviewers invariably have different (new) comments and one can end up with an endless string of revisions. I'm going to cut through that now to force convergence. I will not send out a new (and final) revision for review, but will assess your response to the current reviewer 5 myself. The comments by reviewer 5 seem valuable to me and could lead to further improvement of the manuscript. For that reason, I want to give you the opportunity to respond to the comments and submit a final revision. What follows is the standard letter. I hope to receive your revision soon.

Research ethics:

Use of animals and field studies:

Please submit a copy of your revised paper within three weeks. If we do not hear from you within this time your manuscript will be rejected. If you are unable to meet this deadline please let us know as soon as possible, as we may be able to grant a short extension.

Best wishes,
 Professor Hans Heesterbeek
 Editor, Proceedings B
 mailto: proceedingsb@royalsociety.org

Reviewer(s)' Comments to Author:

Referee: 5

Comments to the Author(s)

Schultz et. al. studied the predictors of immune variation based on a very comprehensive dataset of crossbill immune parameters and a suite of annual cycle traits and environmental covariates across four consecutive summers and between seasons within one year. The authors clearly state their intention to provide a comprehensive descriptive study that highlights the ecological determinants of immune function and this has been done. I particularly like the idea of a two-tiered approach to the data analyses employed by the authors, splitting analyses into an exploratory and a more focused section (see suggestion below about making the second tier more hypothetical). They use the non-parametric random forest models to determine the covariates that merit detailed attention, and linear models to test the effects of selected variables on measured immune parameters. This was aimed at getting to more concise and less parameterised final models, and yet not losing on the benefit of highlighting all possible parameters that may affect immune function. Authors conclude that immune function varies substantially in crossbills among summers in four consecutive years and this they argue is driven largely by food resources (annual cone abundance or cone crop scores). Within the year between seasons, authors found that there was less pronounced variation in immune indices and there was no dominant predictor among the variables they considered.

I find no obvious errors in the writing, but I think that more can be achieved from the data. I believe that the findings will make a significant contribution to the field of ecological immunology. However, I have some concerns (suggestions included):

1. Given the number of predictor variables considered in the analyses and the descriptive approach that the authors have adopted, it is difficult to make a critical assessment of the generality of the manuscript in its current form. I understand the difficulty of making predictions about variation in immune function within a life history context, but this is important for guiding a reader's expectation and can be made under specific assumptions, especially because the authors have measured a lot of factors that can influence immune function.

I recommend that the authors should focus the second tier of their analyses on the opportunity provided by the 'novelty' of their study system i.e. testing the effect of the interaction between reproduction and cone crop scores on immune function. This they stated in lines 26 – 28 (see also line 84-94) but did not focus on: "The red crossbill, *Loxia curvirostra*, is a songbird that can breed opportunistically if conifer seeds are abundant, on both short, cold, and long, warm days, providing an ideal system to investigate interactions between immunity, reproduction, and environmental fluctuations." A study cited by the authors did something similar with year-round breeding common bulbuls *Pycnonotus barbatus* in a seasonally arid tropical environment (reference 23). This system presents an even better opportunity since authors have measured the

resource of interest in the environment. The novelty of the system may not give any advantage if it is not fully utilised.

The effects of cone availability and breeding on immune investment needs to be disentangled in the statistical analyses and shown in the figures and tables, so that the conclusion, "Taken together, our findings suggest that reproductively flexible species (e.g., crossbills) can simultaneously invest in breeding and survival-related processes, which may relate to their ability to exploit abundant food resources" can hold. It needs to be backed up by a clear test of the interaction between breeding status and cone crop scores at the time of measuring immune parameters or some time before if a lag effect is expected (not just the annual average). Cone crop scores and breeding status (CP/BP) and their interaction should be retained in the second tier models predicting immune parameters. In addition, the 'variables warranting further interest' can be included as confounding variables. The first tier analyses should have already eliminated many non-significant variables, allowing the effect of the now fewer 'variables warranting further interest' to be tested and discussed alongside the main focus – disentangling the effects of environment and physiological processes on immune function. This will help authors to condense the discussion and make it more general than it is now. Authors can then elaborate more on the life history implications of seasonal variation in immune function as revealed from their system and the added value of their approach.

2. I cannot find the results of the random forest models anywhere in the manuscript, except a reference Figure S1 in the discussion. Since this was a clear objective of the study, the results should be presented in the result section.

3. Table 1 needs to be split in two (between inter and intra annual variation) and expanded to show the full model results for each immune parameter – there is no way of knowing the variables tested and whether they were significant from that table as it is (one needs to go to the results in the text). The p-value of a model is not very useful for any inference on the effects of predictor variables as far as I know. Also, it is neater to report p-values in two decimal places or as <0.001 at most (see also supplementary information).

4. Figure 1 and 2 are too busy and the captions are not very clear. There is no reference to Fig 1 B anywhere in the text and it is not clear why this is included on the panel. Also, the use of the term response is confusing. I will suggest that authors should outrightly call this immune parameter or something more meaningful. The grey grid lines can be removed from the plots to make the figures clearer. Authors stated in the methods (259-260) that lysis was analysed as a binary outcome (0 or 1) but in Figure 1 and 2 this is presented otherwise. Figure 1 may be moved to supplementary information section – it informs about the method not the results. Figure 2 needs to be split in two (between inter and intra annual variation) and the relationship between immune parameters and cone crop scores needs to be shown in the main manuscript (perhaps something like figure S5).

5. In line 414 – 422, authors highlight a contradictory relationship for haptoglobin concentration: they state that PIT54 was higher in birds in good condition, yet PIT54 was high in the year with smaller cone scores. It is important to highlight why this may arise if higher body condition and cone score reflect quality condition. See

<https://besjournals.onlinelibrary.wiley.com/doi/full/10.1111/1365-2656.13152> for ideas on how similar immune parameters as used in this study may relate differently to nutrition and body mass (condition).

Minor comments

Line 25 – what do authors mean by mild environmental conditions?

It is important to highlight whether all immune assays were carried together or whether samples were analysed differently every year as they were collected. Differences between years may arise from sample handling or storage time, so it is important to highlight that this was considered and

controlled for. The Coefficient of variation of control samples or standards should also be reported if used.

Line 295 – can authors highlight the implication of this significant effect of capture duration and whether it can potentially affect the interpretation of their results?

The title of the supplementary information is different from that of the main manuscript.

What significance threshold for p-value is used for the study? Is there a threshold for R²?

Authors state that "For clarity, models with R² < 0.05 are omitted, and estimates with p-value < 0.1 are bolded." Please, check to be sure that this is right.

Good luck with revisions

Author's Response to Decision Letter for (RSPB-2019-2993.R1)

See Appendix D.

Decision letter (RSPB-2019-2993.R2)

01-Jun-2020

Dear Dr Schultz

I am pleased to inform you that your manuscript entitled "Patterns of annual and seasonal immune investment in a temporal reproductive opportunist" has been accepted for publication in Proceedings B.

Open Access

Paper charges

Sincerely,

Professor Hans Heesterbeek

Associate Editor:

Board Member

Comments to Author:

(There are no comments.)

Appendix A

Response to Reviewers (bold)

Bolded line numbers refer to line numbers in revised manuscript.

Reviewer(s)' Comments to Author:

Referee: 1

Comments to the Author(s)

The manuscript by Schultz et al. describes patterns of constitutive innate immune function among 4 years and among different seasons within one year in Crossbills, a species that has an extraordinary flexibility in their reproductive timing. The authors have apparently invested a lot of time in catching birds and collected a comprehensive dataset. The merit of this study, which distinguishes it from previous work is that the authors also included data on the main food (cone crop) and on weather variables into their analyses. I definitely appreciate this work and the conceptual idea but have a few comments/concerns (in order of importance).

1) The authors used a two-tiered approach for modelling and report results from RFM, overall LMs and test statistics for each parameter within an LM. Then they report what has and has not been included in each model and what was significant and was not. Very often one reads that certain variables had been identified as having high importance in the RFMs, but then eventually most paragraphs within the result section end with phrases like “these variables were not significant predictors in the subsequent LMs”. And indeed if I look at the supplementary tables, then for the analyses among years, only year is significant for all 4 immune parameters but none of the environmental variables. For the within-year (among season) analyses, none of the predictors is significant for lysis, agglutination and WBC. Only for hapotlobin, precipitation was significant. Given those results, I wonder how the authors can conclude that variation in immune function is “most sensitive to environmental fluctuations – specially food resources, precipitation and temperature” (e.g. line 29-30). To me this looks like yes, there is variation between years and that coincides with cone crop but the conclusion that weather has a significant influence seems not correct given the LMs. Finally, the many different types of models give the impression of fishing for some correlated parameters.

Because p-values can be unreliable for linear models with correlated coefficients (please see lines 255-260 in revised manuscript), we have reworked the manuscript to deemphasize variable importance based on coefficient p-values. Instead, we discuss variables with positive importance in the random forest model (variables that improve model predictive power when added) and estimates with $p < 0.2$ in the linear models. This two-tiered approach allows us to identify potential variables we suspect, based on the literature, might affect immune variation using random forest models and then using the more familiar generalized linear model to interpret the covariate effect.

Our conclusion that crossbill immunity fluctuates more in response to environmental variables than physiological variables is supported by the observation that both temperature and precipitation were identified in random forest and linear models

predicting seasonal and yearly variation in multiple immune parameters (Fig. S1, Table S6). In the current manuscript, we have softened the language in lines 32-33 from “is most sensitive to” to “the data suggest that immunity varies seasonally, among years, and in response to environmental fluctuations in food resources, precipitation, and temperature, but less in response to physiological measures such as reproduction”.

We appreciate the reviewers’ concern about “fishing”. We offer in response several key points. First, model selection (and variable selection) in ecology remains an important but often unaddressed aspect of ecological research; our approach is designed with an emphasis on transparency and reproducibility. Here, we started by measuring and including only those covariates that had been identified by published literature as potentially affecting immune variation. We then employ random forest models as a screening tool to identify and remove covariates with no evident predictive power for each immune parameter, a process referred to as variable selection. The subsequent linear models include the same responses and covariates as their associated RFMs. In addition, our results focus predominantly on interpretation of the resulting linear models, including the frequentist inference framework familiar to most ecologists. In this way, we hope to simplify a high-dimensional web of potential interactions into a tractable set of interactions that display consistent importance.

Finally, we hope that our current approach introduces ecologists to the potential benefits of random forest models (non-parametric, can account for non-linear and higher-order interactions), as well as to compare the efficacy of the two approaches.

2) In a high cone crop year there is also lots of breeding, in low cone years is very little breeding. Hence, strictly speaking we cannot tease apart which causes the changes in immune function. It could be either the higher food supply or the reproduction itself, even if the measures of reproduction the authors took did not correlate with immune function because their measure is only qualitative but does not allow any assessment of amount of parental investment (as the authors discuss themselves).

We have added a new figure (Fig. S6) that demonstrates how our reproductive values (cloacal protuberance length and brood patch score) vary annually and seasonally in our study. While we agree that crossbills breed significantly more in higher cone years, crossbills still do breed/come into reproductive condition in lower cone years. As we discuss in the revised manuscript (lines 395-413), our reproductive variables do not assess total reproductive investment, only if the birds were in reproductive condition. Based on Fig. S6, reproductive condition did not vary substantially by cone year. Overall, we are hesitant to say that cone crop and breeding are completely linked--we would need more data in low and high cone years to confirm this--but we have added sentences in the discussion that further explore the limitations of our data sampling (Lines 417-423).

3) Is cone production related to weather variables? Looking at Figure 1 it seems like there might be more (spring) precipitation in a good cone year and fewest precipitation in spring 2012 when the worst cone year followed. If so, then it is difficult to disentangle the effect of weather from food supply.

Based on the literature (e.g., Pearse et al 2014 *Oikos*; Bisi et al 2016 *European Journal of Forest Research*), weather (temperature, precipitation) prior to conifer masting is potentially related to mast/cone crop size, but there is considerable variation among populations/conifer species sampled. We have added this information to Line 334.

4) No birds were caught multiple times, hence we don't know if the variation is due to individual adjustments/adaptations or due to different parts of the (meta)population being present (or catchable). For example, the authors acknowledge that red crossbill abundance fluctuates between years (line 121). Furthermore different vocal types were caught (it remains unclear which vocal types were caught in which year and/or season). This may mean that apparent changes in immune function might be driven by different parts of the population being sampled. Alternatively (and not mutually exclusive), different densities may produce different pressure on the immune system which then would coincide with food supply.

We have added a new table (Table S2) that lists the sample sizes for each vocal type by season and year. In addition, we have added a discussion of this sampling limitation to the discussion section (Lines 423-427).

5) The authors did elaborate statistics with respect to the immune parameters, but only describe the variation in cone crop. Why not presenting statistical test that confirm which is a large cone crop year and which are low crop years?

We have added a new figure (Fig. S4) that displays the expected marginal means of each year's cone crop. Confidence intervals do not overlap or include zero.

6) In general I found the discussion too often just describing the results and the discussion very close to the own results. A broader integration and conceptual perspective would benefit the discussion.

We have substantially reworked the discussion based on reviewer comments. Specifically, we added more discussion of the results using literature from more diverse study systems, reduced some of the crossbill specific sections, and added to the discussion of sampling limitations of the study.

7) The table and Figure captions are not self-explanatory. Please improve them by e.g. mention species name. For Fig 2 it should include that the different seasons have only been measured in one year. For table 1, please make explicit that the data for the seasons are only from one year, etc.

We have added detail to all figures and supplemental tables and figures.

Specific comments

- Line 29: “demonstrate”; you might want to tone down as your data are correlative and you have not done experiments; suggest might be more appropriate

We have made this change (Lines 32-33).

- Line 31: just reading the abstract it is not clear what those physiological factors are. Most people might think of hormones, oxidative stress, telomeres or other physiological systems.

We have added to Line 34: “physiological measures such as reproduction”

- Line 38: change to “most organisms”; species around the equator or in deep oceans don’t experience extensive seasonal variation.

That sentence now reads: “Many temperate, terrestrial organisms experience extensive seasonal variation in weather, disease potential, and resource availability across the annual cycle.” (Lines 41-42)

- Line 60: I think you mean annual-cycle stage here and not life-cycle stage

We have made this change (Line 62).

- Line 86 harsh winter; I suggest to change this to cold winter (to match mild summers); if food is so abundant that birds can breed, the conditions can’t really be harsh

We have changed this sentence to read: “Here, we present a multiannual study of a songbird, the red crossbill (*Loxia curvirostra*), that breeds both in summer and winter if food (i.e. conifer seeds) is sufficiently abundant” (Lines 87-88)

- Line 88: it might be good to briefly mention here already what those environmental and physiological factors are

We have made this change (Lines 89-90).

- Line 112: please add some info on whether one birds makes multiple broods or if the breeding season is long because different individuals breed at different times

We have added that crossbills can have multiple broods /year to Line 114.

- Line 118 and 181: please write Wyoming instead of WY (unless you only expect readers from within the US ;))

We have made this change (Line 120).

- Line 132: here you write that you took 300 ul of blood but later you report that you had to reduce the plasma used for assays because of plasma shortage. Does it mean you used plasma for other studies? No problem with that and I generally appreciate if samples are used for multiple studies/purposes but it might be worth to report that to solve this discrepancy.

Plasma in 2010 was used to optimize the hemolysis-hemagglutination assay. We have added this to lines 208-210.

- Line 162-165: consider moving this part to after line 129. It doesn't really fit under the heading "environmental measures"

We have made this change (Lines 130-135).

- Line 178-179: If you only accessed those weather variable for "each day of bird capture", how could you analyse 1,2,4,8,16 and 32 rolling windows? I guess you mean for bird capture day and those days before.

In addition to accessing weather variables from day-of-capture, we decided post-hoc to compute rolling means for each variable over ecologically relevant time ranges. We have tried to clarify this process in lines 185-194.

- Line 183-184: what is the rationale for taking the temperature difference? **We calculated the temperature difference (Tdiff) for each capture day to compute diurnal temperature range, which can vary substantially in a montane environment (Lines 180-182).**

- Line 206: I am impressed by your low inter-plate variation. Well done!

Thank you!

- Line 207: what is the percentage of zero scores and what is the spread in the non-zero scores?

Distribution of lysis scores: 60.7% zero scores; non-zero scores ranged from 0.4-5. This was added to Line 204.

- Line 225: immune parameters or measures (but not responses); responses relate to immune challenges

We have made this change throughout the manuscript.

- Line 230: you have not mentioned yet that you did measure haematocrit

We describe how we measured hematocrit in lines 143-144.

- Line 238 and 247: what do you mean with response? Immune parameter?

We have changed immune response to parameter throughout manuscript.

- Lines 311-318: this paragraphs reads like a results section. Consider moving it to the results.

In the discussion overall, we have reduced results summarization in favor of more broader context.

- Line 340: this info that the within-year analyses (2011) was done in a large cone crop year might better come earlier (in intro or methods already).

We explicitly detail the timing of data sampling throughout the revised manuscript, but importantly have added that to the abstract and methods sections.

- Line 375: "correlate with", you can not prove causal relationships;

We have removed this language.

- Line 400-401: this seems trivial: if the cost of winter reproduction would not be manageable, then the birds would not breed in winter!

We have changed that sentence to read: “Despite elevated demands, crossbills cope with the high cost of winter reproduction in part because reproduction occurs in winter only if conifer seeds are abundant and foraging efficiency is high, which may allow for investment in immune parameters.” (Lines 407-410).

- Figure 1B: if I understood the data analyses correctly, then two datasets have been used. One for all 4 summers and one for all of 2011. Hence I suggest to remove the winter 2012 data from the figure as those are not part of the analyses. Or do they belong to one winter 2011/12? If they have been treated as one winter in the analyses, then the data should be pooled in the figure as well and not presented separately.

We have removed Winter 2012 and added clarification on sampling timing to the Figure 1 caption.

- Figure 1D: why do you show data from 2014?

We have removed weather data from 2014.

- Figure S3: the axis is in parts unreadable

We have fixed the axis labels for now Figure S1.

- Figure S4: please explain the abbreviations in the figure caption/legend.

We outline the abbreviations in the Figure 1 caption and have added a supplemental table of abbreviations for covariates (Table S3).

Referee: 2

Comments to the Author(s)

The study presented in manuscript RSPB-2018-2450 uses a species with an uncommon natural history to test alternative hypotheses from ecoimmunology about the proximate causes of investment in immune defenses; a beautiful example of the Krogh principle. Specifically, the study tests whether life cycle events (i.e., reproduction, moulting) or environmental traits have a stronger influence on investment in a variety of immune defenses. Disentangling these hypotheses is typically difficult because most species time their annual cycle to correspond with changes in the environment and the effects of the environment and annual cycle events are conflated. This study, however, utilizes red crossbills, which can breed during the summer and winter because they facultatively respond to the abundance of cone crops, their major food source. In short, this manuscript has the potential to advance our understanding of the proximate effects that drive investment in immunology.

The paper is generally clearly written and I appreciate that it includes a section about the limitations of the sampling regime used (lines 388-434). I have a few major suggestions followed by minor comments.

1) The interpretation of the results rests on the assumption that life cycle can be disentangled from environmental traits because red crossbills can breed during both the summer and winter. The manuscript included a matrix of the correlations between the environmental and life cycle traits used in this study, but no evidence is provided that measure of reproduction, moult, or physiological condition do not systematically vary with year or season. It could be that associations between covariates and immune responses are not shown statistically because the main categorical trait (i.e., season or year) is masking the relationship and hiding collinearity between traits. To strengthen the argument that this study disentangles the effects of physiology, life cycle stage, and environment, the lack of an association between time (season and year) and the covariates needs to be shown.

We have added a new supplemental figure that shows the dependence of the continuous physiological covariates (reproductive condition, fat, moult, body condition) on year and season (Fig. S6). While the covariates generally do not vary substantially, there is some annual and seasonal variation among the variables. Thus, we have made note of this limitation in the discussion (Lines 417-423).

2) The discussion is well rooted in the literature about red crossbills. Much research has been conducted on immune defenses in wild animals and in captive, non-model animals, yet little of this research is discussed in the discussion section. I recommend using these studies as context in interpreting the results. For example, the cost of moult are discussed in lines 410-415. Many studies have demonstrated the costs of moult (e.g., Epting 1980 *Physiol Zool*, Murphy 1996 in *Avian energetics and nutritional ecology*) and many have investigated the trade-off between moult and immune defenses (e.g., Moreno et al. 2001 *Oecologia*, Pap et al. 2009 *J Exp Biol*, Ben-Hamo et al. 2017 *J Avian Biol*). These results would provide insights into why it is important to try to capture animals during times of “peak” energy demands. Similarly, published studies might help in interpreting the results about PIT54 (lines 321-331, 363-374).

Thank you for these citation suggestions. We added more discussion of the results using literature from more diverse study systems and reduced some of the crossbill specific sections. In particular, we have expanded our discussion of the relationship between moult and immune function (Lines 378-394) and have cited Moreno et al. 2001, Pap et al. 2009, and Ben-Hamo et al 2017 as suggested.

We have added three new citations to broaden our interpretation of PIT54: Sung 2006; Owen-Ashley & Wingfield 2007; Arsnoe et al. 2017; Downs et al. 2015.

3) The description of the methods for the white blood cell counts indicate that differential counts were performed, yet data for the differential counts are not presented. Including these data might provide additional insights into patterns for the other immune assays given that different types of white blood cells perform different functions and can change differently in response to traits measured (e.g., Matson et al. 2006 Proceedings B).

We examined annual and seasonal variation of eosinophils, heterophils, basophils, lymphocytes, monocytes, heterophil to lymphocyte ratios, in addition to overall leukocyte ratios, which were included in the original manuscript. Eosinophil, heterophil, basophil, and heterophil to lymphocyte ratio models had poor predictive value (very low R^2). Thus, we only report results from overall leukocytes (WBC), lymphocytes and monocytes in the revised manuscript. Seasonal variation in lymphocytes and monocytes were particularly interesting in the context of peak fall moult, which we discuss in Lines 378-394.

4) One of the main conclusions is that the variation in immune defenses seen across years matches with variation seen in cone crop. Cone crop was included as a covariate in the models, but was never a significant predictor of the immune defenses (as far as I can tell), potentially because cone crop is conflated with season and year (see critique 1 above). Similarly, although data about cone index counts are presented, differences across seasons and years were never tested statistically. Please provide these statistics to strengthen the argument presented in lines 311-318. Alternatively, you can perform regression between cone count and the year mean of each immune response to test the relationship presented in lines 311-318.

Cone crop scores for key conifers were only measured once per year. As such, we only include cone year (2010, 2011, 2012, and 2013) as a covariate in our annual models, as the cone year effectively determines the cone crop. All immune variables included cone year as an important predictor of annual variation (Fig. S1; Tables S6-12). To examine how averaged cone crop scores are related to the immune measures, we have added Figure S5, a scatterplot of immune parameter estimated marginal means and cone crop estimated marginal means.

To illustrate that cone crops varied significantly by year, we have added Figure S4, which includes the estimated marginal means of cone score by year with 95% confidence intervals.

5) In figures 1B, it appears that there is only complete data for 2011, but summer data are available for all 4 years of the study. Clarify the sample sizes per season per year by extending table 1 and discuss how these missing data might limit the interpretation of the results. The

imbalance in the design might be hiding important patterns. Similarly, are all of the seasonal effects being driven by 2011?

We have expanded Table 1 (now Table S1) to include sample size by year, season, and immune parameter. We have added clarifying details (season= all seasons, Cone year 2011 only; Cone Year=all years summer only) about the sampling time in captions for Figures 1 and 2, the abstract, methods, results, and discussion section. In addition, we discuss this sampling limitation in the discussion section (Lines 417-423).

Minor comments:

Lines 131-135: Partition this into more than one sentence to improve clarity.

We have split this sentence as requested (Lines 137-139).

Line 206-208: Thank you for including inter-plate variations.

You're welcome!

Lines 325-326: Move the parenthetical so it is after "immune parameters".

We have made this change. (Line 340).

Lines 324-327: Are there other studies that support this idea? Arsnoe et al. 2011 PLoS One and Downs et al. 2015 PLoS One come to mind.

Thank you for these citation suggestions. We have integrated both into the discussion of condition and immunity (Lines 373-377).

Lines 343-347: A complex, confusing sentence as written.

This sentence no longer exists. We have split our discussion of seasonal temperature effects and seasonal precipitation effects into separate paragraphs to improve clarity (349-360; 361-377).

Lines 366-369: Provide a reference for this idea.

We cited Owen-Ashley & Wingfield 2007 (*J Ornithol.*) in support of this idea.

Lines 376-380: What is the number of the reference for Hegemann et al. 2012?

In order to reduce repetition in the discussion, we removed that paragraph containing that reference.

Figure 1B: I think that this figure could be presented in grey-scale if the center symbol (currently a dot) of the cone and bird surveys was changed to a different symbol for one of the items.

We have made Figure 1 into a grey-scale figure.

Supplemental materials: Provide definitions of abbreviations for abbreviations used in each table or figure. The lack of definitions makes it very difficult to interpret the results.

We have detail the abbreviations for immune variables in the figure 1 caption and have included a supplemental table that details covariate abbreviations (Table S3).

Supplemental tables: inconsistent bolding of significant results.

We have fixed this error.

Figure S1: Provide the duration of the window of the rolling means.

We have added the window duration of rolling means to Table S5 (which we reference in now Figure S2 caption).

Figure S2: Fix the overlapping numbers on the x-axis.

We have fixed this issue (now Figure S1).

I couldn't open the file containing the R code and data because my computer didn't recognize the .7z extension.

For review, data files and R code available at Data Dryad

(<https://datadryad.org/review?doi=doi:10.5061/dryad.0fh3jn0>)

Appendix B

Response to reviewers (**in bold**)

Bolded line numbers refer to line numbers in revised manuscript.

Associate Editor Board Member

Comments to Author:

The authors have done a lot of work to address prior concerns by reviewers and have helped clarify many points and have greatly improved the quality and clarity of the manuscript. However, there are a still significant clarifications and concerns with the analytical approaches. The scope of the field sampling is greater than what most current researchers in this field have done, which makes this paper significant. However, the authors are finding little explanatory value in the variables they are investigating. More specifically the most significant issues remaining are 1) the combined use of GLM and RFM still needs further clarification as both myself and the reviewer were unclear as to the approach and interpretations based in these analyses; and 2) I agree with the overreliance on strict 0.05 p values in the field today, however many of the relationships found in the current study were weak (>0.2) and thus some of interpretations seem overly strong based on the statistical findings. In addition, there is also a list of more specific issues raised by the reviewer.

Reviewer(s)' Comments to Author:

Referee: 3

Comments to the Author(s).

GENERAL COMMENTS TO THE AUTHORS:

The study ("Patterns of annual and seasonal immune investment in a temporal reproductive opportunist" explores variation of red crossbills (*Loxia curvirostra*) immune parameters in response to individual and environmental conditions based on bird individuals captured during four years (2010-2013). Birds were mostly captured in summer, but captured in different seasons in one of the study 'cone years' (2011). I agree with the other reviewers that this study uses an interesting model system to explore the timely question of whether immune investment varies in response to environmental conditions, resource availability and individual condition/reproduction. While I also appreciate the considerable efforts the authors have undertaken, I was unfortunately not able to understand all details of the analysis (which I found somewhat inconsistent) and was not convinced by the results. This is because of the following main reasons:

1) The authors used basic Spearman correlation tests to explore any relationships between their immune response variables and climate conditions, according to Table S5, all correlation coefficients were < 0.3 , suggesting weak relationships.

Our statistical approach was as follows: we first used random forest models (RFMs) as a first round of selection to identify variables that demonstrated a consistent statistical association with the immune measures (and removing those variables that did not). Next we constructed linear models (GLM for lysis) using these variables to determine effect sizes and p-values. We make this more explicit in the “Statistical Analysis” section of the manuscript.

In the previous draft, we sought to identify the best time range (window length) over which to average each weather covariate to best predict immunological parameters. In the interest of clarity and simplicity, we have omitted all consideration of moving averages of environmental variables (Table S5 in previous submission). We now consider only values of environmental covariates on the day of bird sampling. We note that this change only affects a subset of seasonal models (e.g., Table 1), particularly “Lysis”, which exhibits a somewhat smaller R^2 in the current version.

2) The authors combine random forest models (RFM) with generalized linear models (GLM) for analyzing the relationships between their immune response variables and suites of covariates, including environmental attributes and measures of individuals attributes of captured birds. The intent to use the RFMs for variable selection and the GLMs for inference based on selected covariates. In terms of the modelling procedure, I found it difficult to understand if this approach was actually necessary but more importantly, I was lost in the results section: some of the statements in the results are based on the relative importance of variables in the RFMs but it is not clear if these are meaningful results at all: if all variables would have little explanatory power, even those with the highest relative importance would be meaningless in explaining variation in the response variables? What I missed in the results were clear statements that covariates had significant effects for explaining variation in the response variables, based on coefficient estimates being clearly distinct from zero or model-based inference (based on information criteria, for GLM, for example).

We have rewritten the Results section to reflect this issue. Specifically, we focus on linear models results. For completeness, we report all effect sizes of continuous predictors (Table S6), and highlight those with $p < 0.1$. In the text, we clearly distinguish between “negative evidence” (variable selection step) and “positive evidence” (statistical significance within familiar LM framework) and emphasize only those variables within the linear models that were consistent, significant ($p < 0.05$) predictors of the immune variables (Lines 265-308).

In addition, we have added a section that highlights which variables were included in the random forest models but excluded from the linear models, and a section that describes variables that were frequently included in

overall significant linear models but were not individually significant within the model (Results, *Variables warranting further study*) (Lines 309-320).

3) The authors state that ‘cone year’ had most impact on variation in their immune measures, which indicates some variation over years. I agree with reviewer 1 that such relationships does not tell anything about immune investment in response to resource availability unless I missed some relevant results? I therefore was not able to find strong support for any conclusions on the role of food resources as stated in line 35 (abstract).

We included cone year first in our random forest analysis and in the subsequent linear model analysis. Cone year was a significant predictor of annual variation in complement, natural antibodies, PIT54, leukocytes/erythrocytes, lymphocytes, and monocytes. Based on this result and previous reviewer feedback, we computed yearly estimated marginal means (EMMs) of cone crops by cone year (marginalized over conifer species) to compare yearly EMMs between each immune variable and cone crop (Fig. S5). We have clarified this approach in the “statistical analysis” section (Lines 261-263) and throughout the results section.

Please apologize if I misinterpreted your approaches and results, in which case I am looking forward to learn from the authors’ responses.

Please find specific comments below (most comments are focused on the methods as I think the results section need a major revision to report clear coefficient effects). Hopefully, some suggestions are useful to the authors.

SPECIFIC COMMENTS:

Lines 30-31 You mention “immune investment” and “immune variation” without going into more details which particular expressions etc have been investigated, and I found this a bit vague.

We have now specified the immune measures used (Lines 29-31).

Line 34 Perhaps consider to mention the particular component of immunity found to exhibit seasonal variation? I found this too vague without reading anything in the abstract about the immunity components explored in this study.

We have tried to make the abstract more specific without being overly detailed given the complexity of the results (Lines 31-36).

Line 36 Please check the expression “physiological measures such as reproduction”: perhaps ‘reproduction’ would be better described as a demographic measure?

We respectfully disagree that reproduction should be considered a demographic measure. While we did not quantify total reproductive

investment, our measures of male cloacal protuberances and female brood patches are fundamentally physiological in nature due to their hormonal control.

Line 48 Perhaps replace “disease potential” with “disease exposure” or “disease susceptibility”?

We have made this change.

Line 68 The term “stages of the annual cycle” is not entirely clear – ‘stages’ of what?

We have added the term “life-history” before annual cycle to clarify (Line 60).

Lines 117-119 I don’t think that the statement “RFMs, however, lack the familiarity and robust inference framework of LMs. As such, we used the variables selected by each RFM to build a corresponding LM” is necessarily true. Can you provide any representative reference for this? If LM are more robust for inference (in your opinion), why don’t you practice variable selection in an LM framework rather than RFM? Suggest to delete this statement, which is also of limited relevance in the introduction.

We clarified our overall statistical approach in lines 98-107.

Line 132 Which “nine month”?

We have changed that sentence to read: “In years with abundant cone crops, crossbills can breed from late summer to the subsequent spring, with a brief hiatus in late fall for moult, and can have multiple broods per year, despite thermal challenges and short days in some seasons” (Lines 113-115).

Lines 221-231 I do not fully understand how your Spearman correlations test were linked to the rest of the analysis (RFM and LMs)? Did you aim to select time windows for averaging weather variables based on strongest correlations? Overall, I think this approach needs some clarification. Moreover, I think it would help to mention all statistical analysis in the paragraph “Statistical Analysis”. Perhaps it will then also become clear why you do all the separate correlation test while also using random forest models as a variable selection tool.

We have removed this statistical approach entirely and now only consider environmental covariates on the day of capture (see above).

Line 222 Which window length(s) were considered ‘plausible’ in your particular study and for your study organisms? Please specify.

We have removed this entirely.

Line 265 Provide the version number of R used in your study.

We have added this to Line 216.

Line 265 Which thresholds was used for “where not highly correlated”? Suggest to state this here.

We have added the correlation value to line 220.

Line 277 Confusing to mention “cone year” in context of analysing samples from summer periods, which do not match “cone years”?

We have removed “cone year” from that section and have added to lines 232-233 that each yearly and seasonal model included sampling period (cone year or season) as a covariate in our models.

Line 283 Please check: while you refer to Table S3 as an overview of “all measured covariates” the legend of Table S3 states “List of covariates selected by random forest models”.

We have added to the caption the list of variables that were removed from the random forest models (Table S3).

Lines 299-307 I do not understand your arguments here and I am confused: if random forest are conducted as a tool for variable selection, why are you discussing variable/model selection for GLMs? Also, there is certainly also literature that emphasise the problems that collinearity can cause in terms of coefficient estimates. Again, why not avoiding any collinearity issues given you multi-step approach? Given the relatively small number of covariates, you may also state which variables were strongly correlated.

We have removed this paragraph.

Lines 306 309 What do you mean by “time period”? This term has not been defined before and is unclear.

We have added clarification for how we define “sampling time” (yearly, seasonally) to lines 229-233.

Lines 310 Please check: should “LM” be replaced by “GLM” (given that you also use logistic regression models)?

We did use a logistic regression GLM to model lysis. However, as noted in the text (Lines 252-253), we refer to this model as the “Lysis LM” for simplicity (as the remaining immune measures were analyzed using linear models).

Legend Table S2 I found the description “Table S2: Summary of observations...” confusing: are you talking about the number of captures or the number of captured individuals. If this table list the number of captures, how did you deal with recaptures in your analysis?

We have changed observations to samples in this caption. There were no recaptures in this dataset and we have added that to the Table S1 caption.

Lines 311 We did you choose “ $p < 0.2$ ”? This value seems to be high given that many studies use a threshold of $p = 0.5$ or $p = 0.01$ for concluding about ‘significance’. I think your value warrants some justification. Also, I suggest to mention the test statistics on which these p-values are based.

We have now included LM regression coefficients for all continuous covariates in Table S6, and bolded those with $p < 0.1$ for clarity. In the text, we focus our discussion on covariates with p-values < 0.05 , and provide a brief additional discussion of non-significant covariates that were included in multiple models (*Variables warranting further study*). In addition, we have clarified in Lines 253-254 that the p-values were calculated using Type II ANOVAs.

Appendix C

We are thankful for the opportunity to revise this manuscript. Our responses to the reviewer are in bold. Bolded line numbers refer to line numbers in revised manuscript.

Referee: 4

Comments to the Author(s).

RSPB-2019-2993 (“Predictors of immune variation”) by Schultz et al. is a generally clearly written manuscript that characterises within and among year variation in a wild passerine. The analysis is extensive, with much material included as supplementary material. For me, the quantity of analyses and materials presented makes a critical evaluation of the discussion (and results) a bit challenging. (And I’ve highlighted examples of apparent conflicts below.) Nevertheless, I think this work is a valuable and comprehensive contribution to the field of ecological immunology/physiology.

One of my bigger concerns in relation to the publication of this manuscript in RSPB is the broader context in which the authors place their work. In its current form, it seems to me that several highly relevant references have not been included in the introduction (and discussion), and one result is that the novelty of the work feels slightly exaggerated. For example, L59-60 states “studies of birds have focussed on single life-history stages in the annual cycle.” This is not true: one can look to the lab of Tielemann, which has examined immune function across annual cycles.

E.g.:

Immune function in a free-living bird varies over the annual cycle, but seasonal patterns differ between years

<https://link.springer.com/article/10.1007/s00442-012-2339-3>

(This paper is cited in other contexts, but it contradicts L-59-60 statement. Same is true of reference 63, Buehler et al.)

But, there are other papers too, e.g.:

Genetic and phenotypically flexible components of seasonal variation in immune function

<https://jeb.biologists.org/content/217/9/1510>

Perhaps, even more relevant is the group’s work on tropical birds that relates to what Schultz et al. refer to as “disentangling the effects of environment and physiological processes on immune investment. Examples of existing work include the following:

Constitutive innate immunity of tropical House Wrens varies with season and reproductive activity

<https://academic.oup.com/auk/article/136/3/ukz029/5486174>

No downregulation of immune function during breeding in two year-round breeding bird species in an equatorial East African environment

<https://onlinelibrary.wiley.com/doi/full/10.1111/jav.02151>

Seasonal differences in baseline innate immune function are better explained by environment than annual cycle stage in a year-round breeding tropical songbird

<https://besjournals.onlinelibrary.wiley.com/doi/pdf/10.1111/1365-2656.12948>

Geographical and temporal variation in environmental conditions affects nestling growth but not immune function in a year-round breeding equatorial lark

<https://frontiersinzoology.biomedcentral.com/articles/10.1186/s12983-017-0213-1>

Incorporating what is already known from these papers (and possibly others) will add important context to the manuscript.

Thank you for pointing out other studies that have looked at variation in immune function across multiple stages in the annual cycle. To address this point, we have amended Line 61 to read “while the majority of studies on birds have focused on single life-history stages of the annual cycle”. We also changed Line 64 to read “Fewer studies, however, have examined modulations in immunity” instead of “few”.

In addition, while we have now highlighted most of these recommended papers in the introduction (Lines 77-83) and discussion (Lines 437-454), we note that none of the papers that the reviewer cited feature systems that breed on both short and long days AND are in environments with extensive variation in food supply, precipitation, and temperature, thus we maintain that highlighting the novelty of our study is warranted.

In the sub-section “Local Weather Conditions” (and L222), I was a bit surprised to read that conditions were only recorded/used in analyses over the 24 hours around capture. It seems unrealistic to me to think that the temperature of the day of capture, and not the week or month (e.g., average) before capture would be important. I would expect possible lag effects here. Can the authors better justify this decision?

In a previous draft of this manuscript, we did discuss possible lag effects of weather variables on the immune variables. Specifically, we determined the best time range (window lengths 1, 4, 8, 16, and 32 days prior to and including day of crossbill capture) over which to average each weather covariate to best predict immunological parameters. However, in response to previous reviewers and in the interest of clarity and simplicity, we omitted all consideration of moving averages of environmental variables. We currently consider only values of environmental covariates on the day of bird sampling. Importantly, these results broadly agree with the corresponding unlagged observations. Substituting the unlagged weather covariates with rolling means did not substantially change LM results and, in the interest of clarity and manuscript length, we do not discuss these results in the current version.

I was also a bit puzzled by the decision to “focus on mean yearly cone crop scores,” when the four vocal types studied each specialise on a different cone type. This seems to be further complicated by the fact that the vocal types are not equally represented in the study. Type 5 was most commonly sampled, and this type prefers Lodgepole Pine and Engelmann Spruce over Douglas Fir and Blue Spruce <https://ebird.org/news/recrtype/>. Would it not make more sense to focus on the preferred resources?

We respectfully disagree with this interpretation of crossbill biology. There are no published data on “foraging preferences” for any North American red crossbill form. Benkman’s (1987) field studies did not discriminate among crossbill types, and in any case were conducted in the Northeastern US and Southeastern Canada, where none of

the trees in question in our paper occur. References that some infer type-specific “preferences” from exclusively measured foraging performance on cones provided to captive birds; the birds were never given choices. Further, these foraging efficiency measurements for different vocal types have been made only on a subset of conifer species and never on blue spruce (and as just noted, only with captive birds, not in the field) (Benkman 1993, Groth 1993, Irwin 2010). We have observed extensively the crossbill types in Jackson Hole foraging on all of the measured conifer tree species (Douglas-fir, blue spruce, lodgepole pine, and Engelmann spruce), including via more than 1500 hours of radiotelemetry-guided observations – which remove the possibility of observer bias in locating and observing different types within different habitats and conifer species. In particular, vocal Type 5 birds use all of the tree species, and especially (and often exclusively) use Douglas-fir and Blue Spruce during peak breeding (e.g., Kelsey 2008, J Cornelius unpublished data). Thus, focusing on the mean yearly cone crops scores of these tree species would accurately reflect food resources for all of the vocal types.

In the discussion, I find a particular passage (L374-381) quite confusing. How/why do the authors propose that birds would use a higher cost defence when food resources are low? Doesn't this go against all trade-off theory? The authors try to reconcile this by explaining that the costs of the 'lower-cost options' (I'm paraphrasing here) are higher than an “expensive inflammatory response.” To me, the reasoning here comes across quite circular and suggests that “high cost” and “low cost” defences have been incorrectly defined/categorised. The bottom line is this text could/should be clarified.

We have clarified the interpretation of these data in lines 376-381: “Higher levels of natural antibodies and acute phase proteins but lower levels of complement and leukocytes during lower cone years may reflect a shift in immune investment strategy in response to lower food resources. While investment in the more rapid but costly PIT54 defence contradicts the energy-limitation hypothesis (Buehler et al. 2009; Owen-Ashley 2007), this trade-off suggests that the overall costs of constitutively maintaining higher levels of protective proteins and cells could be higher than occasionally inducing an expensive inflammatory response.”

Minor comments:

L74: Should this be specifically “inter-annual variation”?

We have made this change.

L165-167: Can more details (perhaps methodological data) be provided to waylay any concerns of readers that having a second observer of one season (summer 2012) did not lead to any systematic differences?

For clarification, we have added to Lines 173-175 that D. Jaul was calibrated against EMS by having her score a subset of the same slides to validate cell ID and quantification.

L171: “Poor predictive value” of what? Can you explain a bit more thoroughly?

These models had poor model-fit ($R^2 < 0.1$). We have added this to Line 179.

L303, L305, L313, L315 (elsewhere?): Perhaps a point of semantics, but I was a bit confused by the use of “predictors of seasonal variation” and “predicting annual variation.” I don't think you are predicting (seasonal/annual) *variation*, you are predicting the magnitude of immune indices using models that include season/year and other explanatory variables.

Thank you for highlighting this point. We have revised the text accordingly.

L400-401 and L406-407: Authors write temperatures were highest in summer and precipitation was higher in winter. It's not clear to me which of these colinear variables (season and climate variable) is driving each of these patterns. With temperature, Fig. 2B suggests it is season, not the temp itself. With precipitation, it seems to be the snow, not the season. L300-301 state no significant effects of season on PIT54, but L410-412 state "higher winter season PIT54 in crossbills. I'm confused. It is unclear to me why this discussion is structured the way it is and emphasises the points that it does.

We apologize for the confusion regarding data interpretation-we have reworked that section to improve clarity (Lines 402-413).

L417-418: Is another possibility not that good condition allowed for PIT54 production?

We have added this possibility to Line 418.

L437-440: Please see comments re: introduction missing relevant references. References are also missing here.

We have cited additional references in this section.

L450-452: This concluding sentence seems to come from out of nowhere. There is little attention in the manuscript to topics of mobility and exploitation of food patches.

We have changed this sentence to remove the reference to mobility and exploitation of food patches. It now reads: "Taken together, our findings suggest that reproductively flexible species (e.g., crossbills) can simultaneously invest in breeding and survival-related processes, which may relate to their ability to exploit abundant food resources."

Appendix D

Response to Reviews from Reviewer 5

We appreciate the editor’s detailed feedback regarding this ongoing review process. We agree that the additional reviews contain valuable feedback, and we welcome the opportunity to further improve this manuscript. Parenthetically, we note that some of the suggestions here appear based on personal preference or convention (e.g., rounding p-values), or do not include a clear justification for the suggested change (e.g., “Figure 2 needs to be split in two”). We attempt to address our specific concerns as they arise, and to provide a clear rationale for our choices.

Our response is bolded. Line numbers indicate lines in the new manuscript.

Reviewer(s)' Comments to Author:

Referee: 5

Comments to the Author(s)

Schultz et. al. studied the predictors of immune variation based on a very comprehensive dataset of crossbill immune parameters and a suite of annual cycle traits and environmental covariates across four consecutive summers and between seasons within one year. The authors clearly state their intension to provide a comprehensive descriptive study that highlights the ecological determinants of immune function and this has been done. I particularly like the idea of a two-tiered approach to the data analyses employed by the authors, splitting analyses into an exploratory and a more focused section (see suggestion below about making the second tier more hypothetical). They use the non-parametric random forest models to determine the covariates that merit detailed attention, and linear models to test the effects of selected variables on measured immune parameters. This was aimed at getting to more concise and less parameterised final models, and yet not losing on the benefit of highlighting all possible parameters that may affect immune function. Authors conclude that immune function varies substantially in crossbills among summers in four consecutive years and this they argue is driven largely by food resources (annual cone abundance or cone crop scores). Within the year between seasons, authors found that there was less pronounced variation in immune indices and there was no dominant predictor among the variables they considered.

I find no obvious errors in the writing, but I think that more can be achieved from the data. I believe that the findings will make a significant contribution to the field of ecological immunology. However, I have some concerns (suggestions included):

1. Given the number of predictor variables considered in the analyses and the descriptive approach that the authors have adopted, it is difficult to make a critical assessment of the generality of the manuscript in its current form. I understand the difficulty of making predictions about variation in immune function within a life history context, but this is important for guiding a reader’s expectation and can be made under specific assumptions, especially because the authors have measured a lot of factors that can influence immune function.

I recommend that the authors should focus the second tier of their analyses on the opportunity provided by the ‘novelty’ of their study system i.e. testing the effect of the interaction between reproduction and cone crop scores on immune function. This they stated in lines 26 – 28 (see also line 84-94) but did not focus on: “The red crossbill, *Loxia curvirostra*, is a songbird that can breed opportunistically if conifer seeds are abundant, on both short, cold, and long, warm days, providing an ideal system to investigate interactions between immunity, reproduction, and environmental fluctuations.” A study cited by the authors did something similar with year-round breeding common bulbuls *Pycnonotus barbatus* in a seasonally arid tropical environment (reference 23). This system presents an even better opportunity since authors have measured the resource of interest in the environment. The novelty of the system may not give any advantage if it is not fully utilised.

The effects of cone availability and breeding on immune investment needs to be disentangled in the statistical analyses and shown in the figures and tables, so that the conclusion, “Taken together, our findings suggest that reproductively flexible species (e.g., crossbills) can simultaneously invest in breeding and survival-related processes, which may relate to their ability to exploit abundant food resources” can hold. It needs to be backed up by a clear test of the interaction between breeding status and cone crop scores at the time of measuring immune parameters or some time before if a lag effect is expected (not just the annual average). Cone crop scores and breeding status (CP/BP) and their interaction should be retained in the second tier models predicting immune parameters. In addition, the ‘variables warranting further interest’ can be included as confounding variables. The first tier analyses should have already eliminated many non-significant variables, allowing the effect of the now fewer ‘variables warranting further interest’ to be tested and discussed alongside the main focus – disentangling the effects of environment and physiological processes on immune function. This will help authors to condense the discussion and make it more general than it is now. Authors can then elaborate more on the life history implications of seasonal variation in immune function as revealed from their system and the added value of their approach.

We agree that testing the interaction between cone crop scores and reproduction would be the best way to truly disentangle environmental and physiological effects on immune function. However, we lack sufficient data to explicitly model the dependence of birds’ immunological parameters on slowly varying cone crop scores. As we note in the methods section (Lines 206-214), cone crops were only sampled annually (between June and September), whereas immune and reproductive measures were sampled seasonally. We do explicitly acknowledge this limitation in the discussion section: “In addition, our characterization of inter- and intra-annual variation in immunity is based on sampling from exclusively summer months and within one cone year, respectively. As such, we cannot fully disentangle environmental and physiological contributors to immunity, particularly because physiological covariates like reproduction, moult, and condition may vary significantly between and within years.”

We have softened the language in the abstract (Lines 26-28) to read: “The red crossbill, *Loxia curvirostra*, is a songbird that can breed opportunistically if conifer seeds are abundant, on both short, cold, and long, warm days, providing an ideal system to investigate environmental and reproductive effects on immunity.” We note that “Variables

warranting further study” (Lines 323-335) section describes covariates that are included in the respective linear models.

2. I cannot find the results of the random forest models anywhere in the manuscript, except a reference Figure S1 in the discussion. Since this was a clear objective of the study, the results should be presented in the result section.

References to Fig. S1 were cut in earlier versions of the manuscript to address previous reviewer comments to make the results section easier to follow, but we have added back references to this figure in the Results section.

3. Table 1 needs to be split in two (between inter and intra annual variation) and expanded to show the full model results for each immune parameter – there is no way of knowing the variables tested and whether they were significant from that table as it is (one needs to go to the results in the text). The p-value of a model is not very useful for any inference on the effects of predictor variables as far as I know. Also, it is neater to report p-values in two decimal places or as <0.001 at most (see also supplementary information).

We agree that marginal model p-values do not provide utility for inference on specific model components, though they can nonetheless provide value in assessing the relevance of the overall model to a particular application, e.g., prediction versus description. As such, we believe that the current table structure (including exact p-values) provides readers with a valuable high-level summary. We note that Tables S4-S12 provide extensive detail regarding full model results that are not feasible to include in main text due to space considerations, and much of which may not be of interest to many readers.

4. Figure 1 and 2 are too busy and the captions are not very clear. There is no reference to Fig 1B anywhere in the text and it is not clear why this is included on the panel.

We have simplified Figure 1 to include only panels A-B and have correspondingly simplified the captions. We believe that Fig. 1B provides a valuable visual overview of the experimental design (e.g., sampling schedule), along with a concise summary of immune parameter data used in this analysis. We have also added references to Fig. 1B in the methods section.

Also, the use of the term response is confusing. I will suggest that authors should outrightly call this immune parameter or something more meaningful.

We have fixed Fig. 1B so that it now reads parameter rather than response.

The grey grid lines can be removed from the plots to make the figures clearer.

We believe that grid lines can assist readers' quantitative interpretations of figures. We have reduced the number of grid lines to reduce visual clutter, but we have not eliminated them.

Authors stated in the methods (259-260) that lysis was analysed as a binary outcome (0 or 1) but in Figure 1 and 2 this is presented otherwise.

We have updated Fig. 1 caption to clarify. In Fig. 2, the y axis shows the estimated marginal mean response based on the linear models. For Lysis (top panel), this is the estimated probability of the binary outcome.

Figure 1 may be moved to supplementary information section – it informs about the method not the results.

We agree that Figs. 1C-D are not needed in the main text, and now include these as supplemental figures. We address the inclusion of (modified) Figs 1A-B above.

Figure 2 needs to be split in two (between inter and intra annual variation) and the relationship between immune parameters and cone crop scores needs to be shown in the main manuscript (perhaps something like figure S5).

We note that the reviewer has not provided a clear justification for why such a split is needed, and we disagree that further dividing Fig. 2 would increase the clarity of presentation. In addition, Figure S5 (now S6) was created in response to a previous reviewer’s request to illustrate the predicted relationship between cone crop scores and immune parameters. As noted above, we do not explicitly model dependence on cone crop, and are careful to avoid drawing overly broad conclusions on these results. While we agree that this relationship is interesting, we disagree that further main text figures significantly add to the manuscript.

5. In line 414 – 422, authors highlight a contradictory relationship for haptoglobin concentration: they state that PIT54 was higher in birds in good condition, yet PIT54 was high in the year with smaller cone scores. It is important to highlight why this may arise if higher body condition and cone score reflect quality condition. See

<https://besjournals.onlinelibrary.wiley.com/doi/full/10.1111/1365-2656.13152> for ideas on how similar immune parameters as used in this study may relate differently to nutrition and body mass (condition).

Thank you for this suggestion. We have cited this paper and added this context to lines 429-432.

Minor comments

Line 25 – what do authors mean by mild environmental conditions?

We have replaced mild with “relatively unchallenging environmental conditions” to further clarify.

It is important to highlight whether all immune assays were carried together or whether samples were analysed differently every year as they were collected. Differences between years may arise from sample handling or storage time, so it is important to highlight that this was considered and controlled for. The Coefficient of variation of control samples or standards should also be reported if used.

We have added more details regarding when the hemolysis-hemagglutination assay samples were run (they were run in five batches) and we cite a recent paper that states that repeat freeze-thaw cycles do not seem to affect the results of this particular assay (Lines 160-165). We have also added the coefficient of variation of the control samples (all control plasma was taken from the same individual rooster) between these plates.

For the haptoglobin assay, we have added that we ran all seven plates on the same day in October 2014. We calculated the average inter and intra CV of one of the standards and reported this in the methods section (Lines 166-174).

Line 295 – can authors highlight the implication of this significant effect of capture duration and whether it can potentially affect the interpretation of their results?

We do discuss the implication of this result in the discussion (Lines 393-395): “In addition, capture duration was negatively related to lymphocytes and overall leukocytes/erythrocytes, suggesting that handling time and thus corticosterone may affect leukocytes [71]”.

The title of the supplementary information is different from that of the main manuscript.
We have made this change.

What significance threshold for p-value is used for the study? Is there a threshold for R^2 ?
Authors state that “For clarity, models with $R^2 < 0.05$ are omitted, and estimates with p-value < 0.1 are bolded.” Please, check to be sure that this is right.

Thank you for identifying this issue. We omit further discussion of model results with $R^2 < 0.1$ throughout and have clarified captions and text. We use a conventional alpha of 0.05 throughout when discussing statistical significance. We do not report exact p-values of model parameters in the text to improve readability of the Results section. Instead, we have added: “please see referenced tables for exact p-values” to the results section to direct the reader. In general, we attempt to avoid strong statements regarding statistical significance based on significance thresholds, and endeavor to report exact p-values so that readers may draw their own conclusions.

For clarity, we have added to the Table 1 caption that we excluded models that lack substantial (and ecologically relevant) overall explanatory power (i.e. $R^2 < 0.1$, also clarified in Lines 183-184). We also changed the caption in Table 1 and S6 to clarify that we focus on models with $R^2 > 0.1$.